# Zebrafish genetic model of neuromuscular degeneration associated with Atrogin-1 expression

Romain Menard[1], Elena Morin[1,2,3], Dexter Morse[1], Caroline Halluin[1], Marko Pende[1], Aissette Baanannou[1], Janelle Grendler[1], Heath Fuqua[1], Jijia Li[1,4], Laetitia Lancelot[1,4], Jessica Drent[1], Frédéric Bonnet[1], Joel H. Graber[1], Prayag Murawala[1,5], Cédric Dray[2,3], Jean-Philippe Pradère[2,3], James A. Coffman[1], Romain Madelaine[1]*

**1** MDI Biological Laboratory, Kathryn W. Davis Center for Regenerative Biology and Aging, Bar Harbor, Maine, United States of America, **2** RESTORE Research Center, INSERM 1301, CNRS 5070, EFS, ENVT, Université Paul Sabatier, Toulouse, France, **3** IHU HealthAge, Toulouse, France, **4** CARe Graduate School, Université Paul Sabatier, Toulouse, France, **5** Department of nephrology and hypertension, Hannover Medical School, Hannover, Germany

* rmadelaine@mdibl.org

## Abstract

The degenerative loss of muscle associated with aging leading to muscular atrophy is called sarcopenia. Currently, practicing regular physical exercise is the only efficient way to delay sarcopenia onset. Identification of therapeutic targets to alleviate the symptoms of aging requires *in vivo* model organisms of accelerated muscle degeneration and atrophy. The zebrafish undergoes aging, with hallmarks including mitochondrial dysfunction, telomere shortening, and accumulation of senescent cells. However, zebrafish age slowly, and no specific zebrafish models of accelerated muscle atrophy associated with molecular events of aging are currently available. We have developed a new genetic tool to efficiently accelerate muscle-fiber degeneration and muscle-tissue atrophy in zebrafish larvae and adults. We used a gain-of-function strategy with a molecule that has been shown to be necessary and sufficient to induce muscle atrophy and a sarcopenia phenotype in mammals: Atrogin-1 (also named Fbxo32). We report the generation, validation, and characterization of a zebrafish genetic model of accelerated neuromuscular atrophy, the atrofish. We demonstrated that Atrogin-1 expression specifically in skeletal muscle tissue induces a muscle atrophic phenotype associated with locomotion dysfunction in both larvae and adult fish. We identified degradation of the myosin light chain as an event occurring prior to muscle-fiber degeneration. Biological processes associated with muscle aging such as proteolysis, inflammation, stress response, extracellular matrix (ECM) remodeling, and apoptosis are upregulated in the atrofish. Surprisingly, we observed a strong correlation between muscle-fiber degeneration and reduced numbers of neuromuscular junctions in the peripheral nervous system, as well as neuronal cell bodies in the spinal cord, suggesting that muscle atrophy could underlie

---

**Data availability statement:** Data availability. All data, reagents, and genetic tools presented in this manuscript are available to the scientific community, either in a public repository, within the manuscript itself or as supplementary information. The transcriptomics data are publicly available at the GEO database and the accession numbers are GSE274353, GSE80221 and GSE144885.

**Funding:** o This study was supported by Institutional Development Awards (IDeA) from the National Institute of General Medical Sciences of the National Institutes of Health under grant numbers P20GM103423 and P20GM104318 (MDIBL), as well as P20GM144265 and 3P20GM144265-01A1S1 (RM). Additional funding was provided by NIH (R03-HD099468) and Morris Discovery Fund awards to JAC. The National Science Foundation REU program DBI-2243416 supported JD. This work was also funded by grants from the DFG (527098031) and National Institute of Health (ORIP-R21OD031971) awarded to PM. EM, CD and JPP are supported by ANR grant (ANR-21-CE14-0054-02), the Inspire Program (Region Occitanie, France - Reference number:1901175) and Fondation pour la recherche médicale (DEQ20180339226). The IHU HealthAge was supported by the French National Research Agency (ANR) as part of the France 2030 program (reference number: ANR 23 IAHU 0011). The funders had no role in study design, data collection and analysis, decision to publish, or preparation of the manuscript.

**Competing interests:** The authors have declared that no competing interest exists.

a neurodegenerative phenotype in the central nervous system. Finally, while atrofish larvae can recover locomotive functions, adult atrofish have impaired regenerative capacities, as is observed in mammals during muscle aging. In the future, the atrofish could serve as a platform for testing molecules aimed at treating or alleviating the symptoms of muscle aging, thereby opening new therapeutic avenues in the fight against sarcopenia.

## Author summary

As people grow older, muscles become weaker and dysfunctional, making everyday activities challenging. The gradual loss of muscle strength and mass is called sarcopenia, the age associated muscle atrophy, leading to frailty and increased risk of mortality. Today, the best way to slow the onset of sarcopenia is through regular physical exercise, but scientists are actively searching for new treatments that could one day help millions of people to live a longer healthy life.

To reach this goal we have modeled age associated muscle atrophy in zebrafish, an animal model sharing many genetic and physiological similarities with humans. In this study, we report the development, validation and functional characterization of a new zebrafish genetic model of muscle atrophy, the "atrofish" By expressing a molecule called Atrogin-1, already known to trigger muscle wasting in mammals, we found that this model of muscle aging develops muscle degeneration and swimming dysfunction early in life, but also show problems in their neurons and nerve connections in the central and peripheral nervous system respectively. This zebrafish genetic model could help uncover new ways to fight muscle aging and sarcopenia to keep people moving and healthy late in life.

## Introduction

Progress in biomedicine has allowed people to live longer; however, living longer does not always mean living a healthier life, as elderly people often suffer from age-related diseases. Such diseases include muscular atrophy, which is characterized by reduced muscle mass and function. The age-related degenerative loss of muscle that leads to muscular atrophy is called sarcopenia. Sarcopenia was recently recognized as a disease by the World Health Organization (WHO) [1]. It affects more than 60% of people over 80 years of age and results in mobility disorders and significantly increased risk of mortality. The etiology of sarcopenia is multifactorial, and its phenotype has been linked to mitochondrial dysfunction, chronic inflammation, cellular senescence, motoneuron degeneration, and a decline in muscle regenerative capacity [2–5].

Currently, the primary means of limiting the decline of skeletal muscle mass and delaying muscle atrophy onset is to engage in regular physical exercise. In healthy

people, muscular activity increases muscle mass, and muscle injury activates the muscle stem cell population and a regenerative response [6]. Unfortunately, these hypertrophic and regenerative responses to physical training and muscle injury, respectively, are extremely limited in the elderly or under certain muscle dystrophy conditions [7]. Given the increasing age of the population worldwide and a projected life expectancy reaching ~78 years in 2050, there is a critical need to identify and characterize potential novel therapeutic candidates that will impede sarcopenia development in elderly individuals who lack the capacity for physical exercise or have inefficient hypertrophic and/or regenerative responses. Achieving this goal requires a better understanding of the molecular and cellular mechanisms underlying age-related muscle degeneration.

In turn, advances in research on muscle aging and identification of therapeutic targets to suppress the symptoms of aging require *in vivo* model organisms of accelerated muscle atrophy and degeneration. The zebrafish has become a widely used animal model for functional studies relevant to human muscle biology and disease [8,9]. Their advantages as a research platform include their small size; their transparency, which facilitates *in vivo* imaging; and the ease of genetic manipulation. The zebrafish has been used during the last 25 years as a valuable model system to study embryonic development and cellular regeneration. In addition, recently the zebrafish has been widely used as a model for studying the molecular and cellular mechanisms underlying aging [10–14]. Zebrafish, which have an average lifespan of approximately three years, harbor hallmarks of aging, notably mitochondrial dysfunction, telomere shortening, and protein oxidation. Also, as in mammals, aged zebrafish exhibit senescent-cell accumulation in various tissues. In addition, skeletal-muscle tissue of old (24-month-old) zebrafish shows symptoms of sarcopenia, including scoliosis, muscle mass reduction, and a decline in muscle regenerative capacity [13,15,16].

While natural aging is physiologically the most relevant model for studying age-related phenomena, development in zebrafish of a severe form of muscle atrophy resembling a sarcopenia phenotype usually requires at least three years, hindering time- and cost-efficient analyses. Although a zebrafish model of accelerated aging has been used to study telomerase-dependent aging, i.e., a mutant for the *tert* gene [10,12,15,16], which codes for a component of telomerase, this mutant may not be the proper model to study aging-associated muscle atrophy. Specifically, while muscle mass is reduced globally in the *tert* mutant, skeletal muscles, surprisingly, do not show increased senescence. This observation contrasts with induced senescence in other tissues of this mutant, suggesting that different telomerase-dependent mechanisms of aging may be involved in replicative versus non-replicative tissues such as skeletal muscles. It has been hypothesized that the sarcopenia-like phenotype in *tert* zebrafish could result from high levels of reactive oxygen species in muscles, the senescence-associated secretory phenotype observed in non-muscle tissues such as the gut, and/or poor nutritional state [12,14,16]. While zebrafish mutants and transgenics have been used to study myopathies and muscular dystrophies, no zebrafish models specific for the study of muscle atrophy and degeneration are currently available to study age-related loss of muscle mass and function [11]. Given the many advantages of zebrafish as a model organism, a zebrafish model of accelerated muscle atrophy and degeneration would be an extremely valuable resource.

As noted above, development of a disease resembling human sarcopenia in naturally aging zebrafish requires at least three years. Here, we report the generation, validation, and characterization of a zebrafish genetic model of accelerated muscle atrophy and degeneration that overcomes some of these limitations. This transgenic zebrafish, which we have named the atrofish, exhibits accelerated muscle-fiber degeneration and muscle-tissue atrophy. We developed this novel research tool using a gain-of-function strategy to overexpress a molecule previously shown to be necessary and sufficient to induce muscle atrophy and a sarcopenia phenotype in mice: Atrogin-1 (Atro1) [17–19]. Atrogin-1 is an E3 ubiquitin ligase involved in protein degradation, and dysregulation of its expression strongly affects muscle-tissue homeostasis [20]. Atrogin-1 shows increased expression in muscles during aging and is involved in sarcopenia onset [17–19], and has been identified as a genetic modifier of Duchenne muscular dystrophy in zebrafish [21]. Interestingly, both *atrogin-1* loss of function [21] and gain of function [17–19] lead to muscle-fiber degeneration, highlighting the delicately balanced regulation of proteolysis required to maintain muscle-tissue integrity. We hypothesized that recapitulating a molecular event that

occurs during natural aging at the onset of sarcopenia, such as Atrogin-1 expression, would be an efficient and biologically relevant strategy for inducing muscle atrophy and degeneration.

We demonstrated that overexpression of Atrogin-1 specifically in skeletal-muscle tissue induces muscle-fiber degeneration and an atrophic phenotype associated with locomotion dysfunction in both larvae and adult zebrafish. In addition, analysis of gene expression by RNA-sequencing indicated that biological processes associated with muscle aging and/or degeneration such as proteolysis, inflammation, and the stress response are upregulated in the atrofish. We also identified myosin light-chain degradation as an event occurring prior to muscle-fiber degeneration in the atrofish. Further, we observed a strong correlation between muscle-fiber degeneration and reduced density of neuromuscular (NMJ) junctions in the peripheral nervous system, and also, surprisingly, degeneration of neuronal cell bodies in the spinal cord. This result raises the intriguing hypothesis that muscle atrophy could underlie a neurodegenerative phenotype in the central nervous system. We also demonstrated that locomotion impairment, neuromuscular degeneration, and associated phenotypes correlated with Atrogin-1 overexpression, are reversible in atrofish larvae. Finally, we validated the efficiency of this new genetic model to induce early muscle atrophy and degeneration in adult zebrafish and showed that reduced swimming performance correlated with this phenotype is not reversible. This observation suggests that the muscle regenerative process is impaired after chronic induction of Atrogin-1 expression and muscle degeneration, and supports the use of the atrofish as a model of accelerated muscle aging in young adult zebrafish.

In future studies, the atrofish could open new therapeutic avenues in the fight against neuromuscular degeneration or sarcopenia and be a valuable resource for the scientific community studying the molecular and cellular mechanisms of muscle atrophy, degeneration, and aging. In addition, the atrofish will be a cost-effective and efficient research platform for *in vivo* testing of therapeutic molecules designed to limit neuromuscular degeneration or promote tissue regeneration.

## Results and discussion

### Atrogin-1 expression in skeletal muscles induces muscle atrophy, degeneration, and dysfunction in zebrafish larvae

To recapitulate the overexpression of Atrogin-1 occurring during the onset of age-related muscle atrophy, we generated a set of unique genetic tools. First, we used Tol2 transgenesis [22] to establish two novel Cre/*lox* lines, i.e., *Tg(ubi:Lox-stop-Lox-atrogin1)* and *Tg(503unc:creERT2)*, to express the Atrogin-1 protein specifically in skeletal-muscle tissue of zebrafish larvae or young adult zebrafish (*503unc* is a small, muscle-specific promoter [23]; *ubi* is an ubiquitous promoter [24]). We then validated the capacity of the *Tg(503unc:creERT2)* transgene to induce *loxP*-site recombination after addition of hydroxytamoxifen (4-OHT) and activation of the inducible Cre recombinase [25]. Results demonstrate that after 72 hours of 4-OHT treatment at 10μM, double-transgenic larvae, i.e., *Tg(ubi:Lox-egfp-Lox-mcherry)* [24] and *Tg(503unc:creERT2)*, exhibit mCherry expression in trunk skeletal-muscle fibers at 6 days post-fertilization (dpf) (S1A in S1 Fig). This observation indicates that the creERT2 protein expressed under regulation of the *503unc* promoter induced efficient *loxP*-site recombination after 4-OHT treatment. We hypothesized that the use of the same strategy in *Tg(ubi:Lox-stop-Lox-atrogin1)* and *Tg(503unc:creERT2)* double-transgenic larvae (these double-transgenic fish are referred to hereafter as atrofish) would result in induction of muscle atrophy and muscle-fiber degeneration following Atrogin-1 expression. Using hybridization chain reaction fluorescent *in situ* (HCR RNA-FISH), we showed an increase in *atrogin-1* expression in trunk skeletal-muscle tissue after 48 hours of 4-OHT treatment (Fig 1A). Finally, following treatment of zebrafish larvae with 10μM 4-OHT for 72 hours, we observed an absence of muscle fibers in atrofish larvae at 6 dpf using birefringence analysis with polarized light, while the organization of skeletal muscles in control larvae (DMSO-treated atrofish larvae) is unaffected, with repeated series of v-shaped myotomes (Fig 1A).

To better characterize the effect of Atrogin-1 expression on trunk skeletal-muscle tissue homeostasis, we quantified the penetrance of the phenotype in atrofish larvae using phalloidin staining to label muscle fibers. We observed multiple

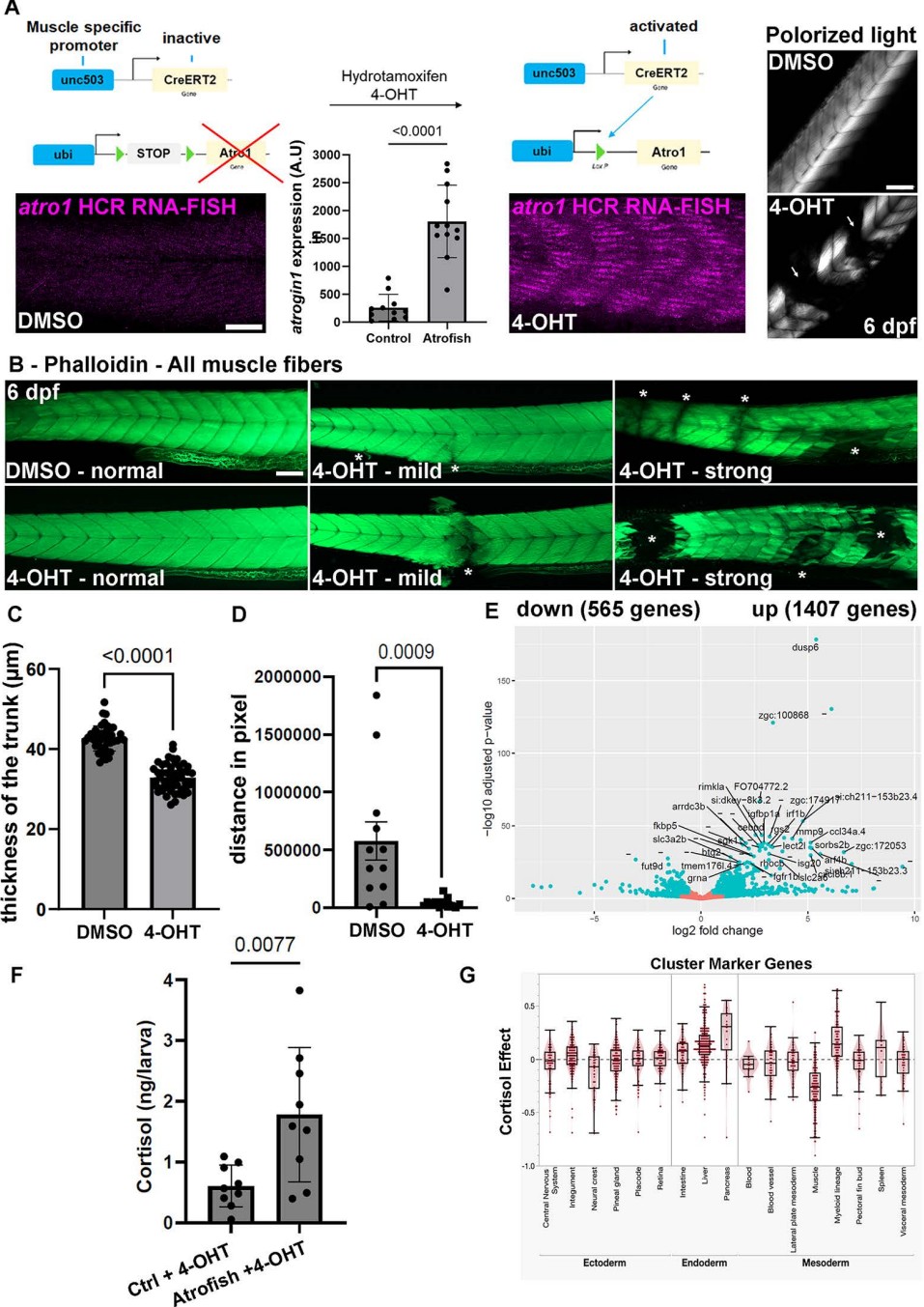

**Fig 1. Muscle-specific Atrogin-1 expression leads to muscle atrophy and affects locomotor function in zebrafish larvae. (A)** Schematic representation of the method used to express Atrogin-1 in muscle fibers using Tg(*503unc:creERT2*) and Tg(*ubi:Lox-stop-Lox-atrogin1*) transgenic lines (atrofish larvae). Efficient recombination of the *loxP* sites following 4-OHT treatment leads to Atrogin-1-overexpression-dependent muscle atrophy and degeneration, visualized by birefringence analysis using polarized light. Statistical significance is determined Mann-Whitney test. **(B)** Phalloidin incorporation in control and atrofish larvae after 72 hours of 4-OHT treatment reveals degenerative phenotypes in trunk skeletal-muscle tissue. Asterisk shows a site of muscle-fiber degeneration. **(C)** Quantification of muscle thickness in control (DMSO; n = 43) and atrofish (4-OHT; n = 51) larvae after 24 hours of 4-OHT treatment. Statistical significance is determined by t-test, two-tailed, unpaired. **(D)** Quantification of locomotor activity in control (DMSO; n = 12) and atrofish (4-OHT; n = 12) larvae for one hour after 24 hours of 4-OHT treatment. Statistical significance is determined by a Mann-Whitney test, two-tailed, unpaired. **(E)** Graphical representation of bulk RNAseq analysis and variation in gene-expression profiles in control (DMSO-treated) (n = 4; 50

larvae/sample) vs. atrofish (n = 4; 50 larvae/sample) after 24 hours of 4-OHT treatment. **(F)** Quantification of cortisol expression levels in control larvae (4-OHT-treated; n = 8 with 30 larvae/sample) and 4-OHT-treated atrofish larvae (n = 8 with 30 larvae/sample) larvae. Statistical significance is determined by t-test, two-tailed, unpaired. **(G)** Graphical representation of RNAseq analysis and variation in the cell populations based on gene-expression profiles in control vs. cortisol-treated. To map the effects of cortisol exposure to cell and tissue type, marker genes and cell types were identified within the clusters by matching the annotations from the full embryo atlas to the 5 dpf single-cell data (see method for details). Each differentially expressed gene is represented as a dot within the cluster. Error bars represent s.d. Scale bars: 100 μm.

phenotypes associated with *Atrogin-1* expression, ranging from normal phalloidin incorporation in muscle fibers (similar to controls: 32.35%); to mild defects, with muscle-fiber degeneration occurring in one site within the trunk (only one site of degeneration: 32.35%); to a strong atrophic phenotype, with large areas of muscle tissue missing in different regions along the dorso-ventral axis of the trunk (multiple sites of degeneration: 35.3%) (Fig 1B; 4-OHT at 10μM for 72 hours). Reduced duration of 4-OHT treatment correlates with a gradual reduction of the phalloidin-incorporation percentages associated with the mild and strong phenotypes; results show only 20.7% and 3.4% for the mild and strong phenotypes, respectively, after 24 hours of CreERT2 activation and Atrogin-1 expression (S1B in S1 Fig). In addition, after 24 hours of 4-OHT treatment between 5 and 6 dpf, we observed an overall reduction of trunk-muscle thickness by ~23% (Fig 1C), supporting the hypothesis of reduced muscle mass in atrofish larvae. Finally, to correlate muscle-fiber degeneration and muscle atrophy with reduced muscle function, we performed locomotion assays. Interestingly, we observed that after only one day of 4-OHT treatment, atrofish larvae show significantly reduced locomotor activity compared to DMSO-only-treated larvae carrying the *Tg(ubi:Lox-stop-Lox-atrogin1)* and *Tg(503unc:creERT2)* transgenes (Fig 1D), indicating that Atrogin-1 expression not only affects muscle-fiber survival but also underlies locomotion defects in zebrafish larvae.

Together, these results demonstrate that Atrogin-1 expression in zebrafish skeletal-muscle tissue is an efficient strategy to induce muscle-fiber degeneration and muscle dysfunction. These data support the atrofish as a potentially valuable genetic model for investigating the molecular and cellular mechanisms of muscle atrophy and associated locomotor dysfunction.

### Molecular pathways associated with muscle atrophy, degeneration, and aging are dysregulated in the atrofish.

To better characterize the molecular mechanisms involved in the atrophic phenotype, we performed a bulk RNA-sequencing gene-expression analysis during muscle degeneration in atrofish larvae. This analysis was conducted at 6 dpf using mRNA from double-transgenic larvae treated with either DMSO or 4-OHT between 5 and 6 dpf. We identified a total of 1,972 genes that are significantly dysregulated following Atrogin-1 expression between 5 and 6 dpf. Among them, 1,407 genes are upregulated while 565 are downregulated (Fig 1E and S1 Table). Gene ontology analysis indicates that response to stimuli, inflammation, extra-cellular matrix composition, and endopeptidase activity are among the main biological processes dysregulated after Atrogin-1 expression at 6dpf (S1C in S1 Fig). KEGG pathway analysis supports this observation, with NOD-like-receptor and cytokine-cytokine-receptor signaling as the two most highly dysregulated pathways (S1D in S1 Fig). Interestingly, components of the apoptosis and p53 pathways are also highly represented among the dysregulated pathways, suggesting that apoptosis at least partially underlies the degenerative process in the atrofish.

We found that genes associated with immune response and inflammation including, but not limited to, *irg1*, *ccl34*, *cxcl8*, *cxcl18*, *mmp9*, and *il1b*, are highly upregulated following Atrogin-1 expression (S1C in S1 Fig and S1 Table). We quantified the total number of macrophages in trunk skeletal-muscle tissue using the *Tg(mpeg1:mCherry)* transgenic line and showed that the total number is similar in control and atrofish larvae, suggesting that increased cytokine expression is mediated by a change in the transcriptome profile of resident macrophages (S2A in S2 Fig). We also identified a transcript named *zgc:100868* that is overexpressed tenfold in atrofish larvae (S1 Table). Interestingly, this gene is predicted to code for a protein with serine-type endopeptidase activity, orthologous to the two human serine proteases PRSS27 and PRSS53, which are known to have proteolytic functions [26]. This observation supports the hypothesis that proteolysis

homeostasis is affected in the atrofish. Finally, sixteen collagen genes are highly downregulated, including *col9a1*, *col1a1*, *col4a6*, *col12a1*, *col2a1* and *col11a1* (S1 Table), indicating that in our atrofish model, ECM composition is strongly affected (S1C, S1D in S1 Fig) by muscle degeneration and associated biological processes. To determine whether these ECM modifications are associated with cellular processes underlying muscle aging such as fat infiltration, gliosis, or fibrosis, we analyzed these mechanisms and demonstrated that fat deposition, and fibrosis are not affected in the skeletal neuromuscular system after *atrogin-1* induction (S2B-S2E in S2 Fig). This observation suggests that variations in ECM composition are mediated by a switch in collagen-gene expression in muscle tissue.

Our transcriptomic analysis also revealed that expression of the endogenous *atrogin-1* gene is increased approximately fourfold following Atrogin-1 expression in atrofish larvae (S1 Table), suggesting a positive feedback loop regulating Atrogin-1 expression during muscle atrophy and degeneration. This observation is consistent with our data showing that FOXO signaling is upregulated in atrofish larvae (S1D in S1 Fig and S1 Table), and with previous evidence that Atrogin-1 is a target gene of the FOXO signaling pathway [27,28]. In addition, a recent study in mice indicated that FOXO transcription factors cooperate with C/EBPδ to control Atrogin-1 expression [29]. Interestingly, a zebrafish ortholog of C/EBPδ, *cebpd*, is highly upregulated in our transcriptomic analysis (S1 Table), suggesting that it may participate in regulation of Atrogin-1 expression. Finally, our RNAseq study also shows that core components of the stress response involving cortisol are upregulated in atrofish larvae (S1 Table). Indeed, known glucocorticoid-induced targets including *klf9*, *fkpb5*, *sgk1*, and *tsc22d3* are highly expressed after Atrogin-1 expression. Supporting this observation, the *pomca* gene, which codes for the precursor of adrenocorticotropic hormone (ACTH), is also overexpressed.

Because inflammation and ACTH control cortisol expression, we next quantified cortisol levels via ELISA and determined that cortisol expression levels are significantly higher in atrofish larvae than in DMSO-treated control larvae (Fig 1F). Interestingly, previous studies have associated high cortisol levels with upregulation of *atrogin-1* expression and demonstrated that this induction is dependent on glucocorticoid receptors function [30–32]. A gene expression analysis to define cell populations affected by chronic cortisol treatment showed that cortisol-treated larvae exhibit a severe negative impact on muscle cells homeostasis (Fig 1G). These data are highly relevant for understanding the etiology of age-related muscle atrophy, because high cortisol levels are associated with muscle atrophy [33–35] and have been suggested as a biomarker of sarcopenia [36]. Together, these observations suggest crosstalk between Atrogin-1-dependent proteolysis, inflammation, stress responses, and apoptosis, potentially leading to neuromuscular degenerative phenotypes and ECM remodeling in the atrofish.

## Myosin light-chain degradation precedes muscle degeneration in the atrofish

The above experiments validated the atrofish as a genetic model of muscle atrophy and reduced muscle function. We next aimed to use it to identify new molecular and/or cellular mechanisms leading to muscle weakness and/or muscle degeneration. Because Atrogin-1 is an E3 ubiquitin ligase, it can potentially target a wide variety of proteins for degradation. While most Atrogin-1 targets are not known, some have been identified, including, but not limited to, calcineurin, aquaporin 4, nuclear factor-kB, MyoD, and cardiac myosin-binding protein C (cMyBP-C) [37–40]. Recently, an elegant report indicated that the endoplasmic reticulum chaperone BiP is also a target of Atrogin-1 contributing to Duchenne muscular dystrophy [21]. Based on the evidence that cMyBP-C is a target of Atrogin-1, we hypothesized that reduced levels of myosin and/or related proteins involved in myosin-complex formation and function could be, at least partially, responsible for the atrophic phenotype observed in atrofish larvae. Using an F310 antibody that recognizes myosin light chains in fast-twitch muscle fibers, we observed dramatically reduced F310 staining in atrofish larvae after 72 hours of 4-OHT treatment at 10μM (S3A in S3 Fig). Darkfield imaging revealed that while F310 staining is absent in the trunk of 4-OHT-treated larvae, the shape of skin, notochord, and myotomes can be observed (S3A, S3B in S3 Fig), indicating that the axis of the larva is not broken but that no myosin light-chain proteins are present in the tissue. Surprisingly, reduced F310 staining is not always associated with the absence of all muscle fibers. In addition, in atrofish larvae we observed retention of

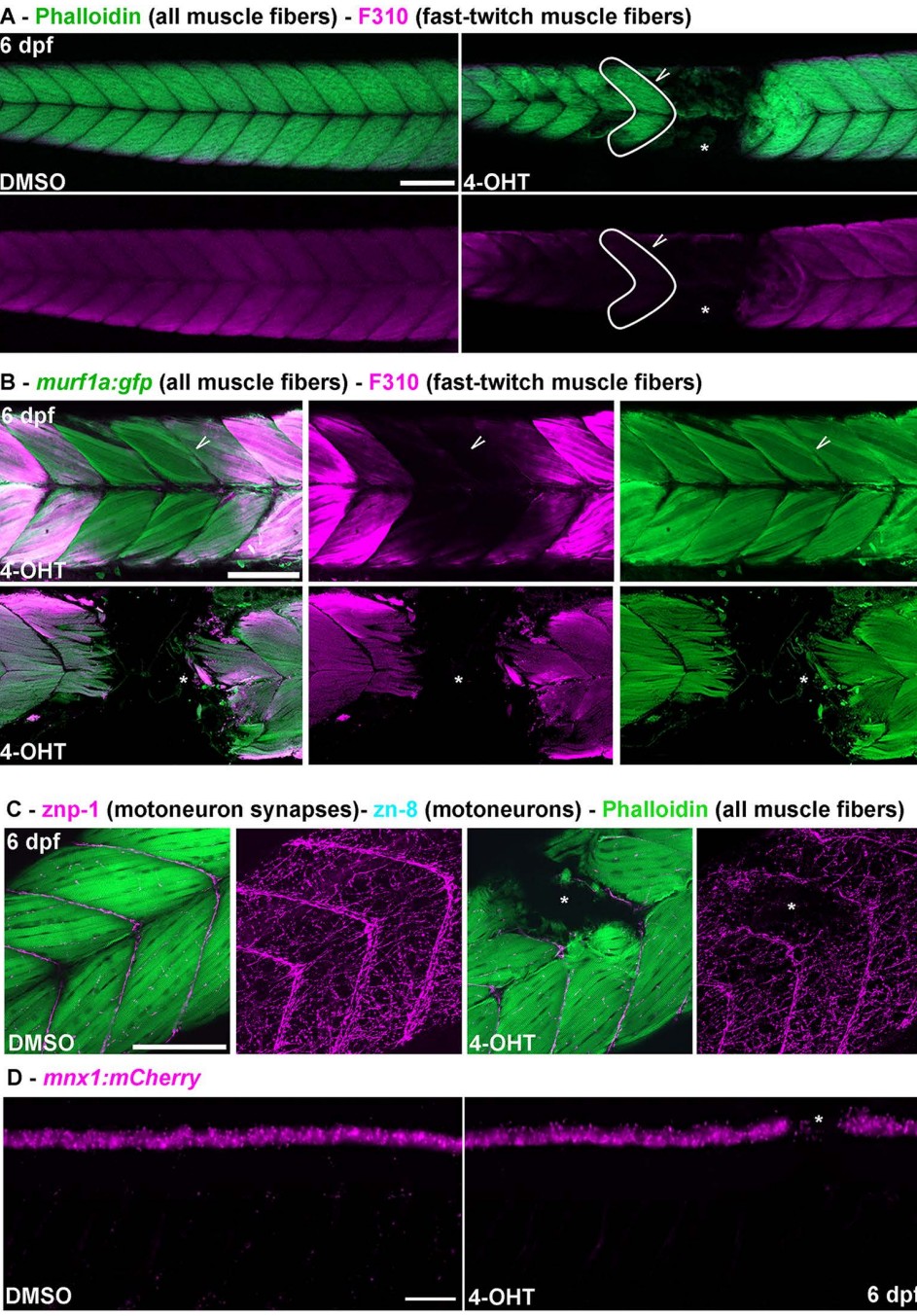

**Fig 2. Muscle-specific Atrogin-1 expression leads to myosin light-chain degradation and neuromuscular degeneration in atrofish larvae. (A)** Apotome sections of DMSO-treated atrofish larvae (left) or atrofish larvae treated with 4-OHT for 24 hours (right) at 6 dpf showing phalloidin incorporation (green) and F310 immunolabelling (purple) in trunk skeletal muscles. White arrow shows phalloidin incorporation in the absence of F310 expression. **(B)** Confocal sections of atrofish larvae treated with 4-OHT for 24 (top) or 48 (bottom) hours at 6 dpf showing GFP expression from Tg(*murf1a:gfp*) in muscle fibers (green) and F310 immunolabelling (purple) in trunk skeletal muscles. White arrow shows GFP expression in the absence of F310 expression. **(C)** Confocal sections of DMSO-treated atrofish larvae (left) or atrofish larvae treated with 4-OHT for 24 hours (right) at 6 dpf showing phalloidin incorporation (green) and znp-1 (purple) immunolabelling in trunk skeletal muscles. **(D)** Apotome sections of DMSO-treated atrofish larvae (left) or atrofish larvae treated with 4-OHT for 24 hours (right) at 6 dpf showing motoneuron degeneration using the *mnx1:mCherry* transgenic line. Asterisk indicates the site of degeneration in the spinal cord. Asterisks show sites of muscle-fiber degeneration. Scale bars: 100 µm.

phalloidin staining in some regions of myotomes in which F310 labelling is strongly reduced (Fig 2A), indicating the presence of muscle fibers despite the absence of myosin light-chain expression. This analysis suggests that myosin light-chain protein can be lost after *atrogin-1* expression and before muscle-fiber degeneration. Quantification of phalloidin and F310 staining over time in atrofish larvae revealed that most larvae exhibiting reduced F310 staining in regions of trunk-muscle tissue also show reduced phalloidin incorporation in those regions (S3C in S3 Fig). Indeed, only ~3% and ~8% of atrofish larvae show reduced or absent F310 staining without an effect on phalloidin incorporation in muscle fibers after 48 or 24 hours of 4-OHT treatment, respectively. This observation suggests that phalloidin incorporation is rapidly reduced when myosin light-chain expression and function are impaired. Finally, co-immunolabeling of atrofish larvae with phalloidin, F310 (myosin light chains in fast-twitch muscle fibers), and A4.1025 (myosin heavy chains in slow- and fast-twitch muscle fibers) antibodies followed by confocal imaging of myotomes after 24 hours of 4-OHT treatment indicated that myosin light-chain, but not myosin heavy-chain, expression can be rapidly reduced independently of muscle-fiber degeneration (S3D in S3 Fig). Together, these results indicate that Atrogin-1 overexpression in zebrafish affects the expression and/or stability of myosin light chains before muscle-fiber degeneration.

To confirm the role of Atrogin-1 in myosin light-chain degradation in the observed atrophic phenotype in the atrofish, we used the *Tg(MuRF1a:GFP)* line, which labels muscle fibers, and also used the F310 antibody to specifically label fast-twitch fibers. As shown with phalloidin incorporation (Fig 2A), we demonstrated that after 48 hours of 4-OHT treatment, areas of myotomes showing reduced F310 staining generally also show an absence of GFP expression (Fig 2B), supporting the notion that the muscle-fiber-degeneration and/or muscle-breakdown phenotype is induced by Atrogin-1 expression. Interestingly, 24 hours of 4-OHT treatment can lead to a reduction or absence of myosin light-chain expression despite the presence of GFP-expressing fast-twitch muscle fibers (Fig 2B). This result confirms our previous observations and suggests that early reduction of myosin light-chain expression primes muscle tissue for degeneration, leading to muscle weakness and dysfunction. In the future, a more precise determination of the timing of myosin light-chain degradation relative to that of muscle-fiber degeneration will be needed to fully understand the relationship between the two processes. However, such an investigation would likely require a destabilized fluorescent protein fused to the myosin light-chain protein to perform live imaging of these degenerative processes. Based on these observations, together, we can confidently propose that this genetic model of myosin complex degradation will be valuable for future identification of therapeutic compounds that can stabilize or prevent degradation of myosin light chains as well as associated proteins during muscle atrophy.

## Muscle atrophy leads to spinal-neuron degeneration in the atrofish

Having identified an atrophic and degenerative muscle phenotype associated with locomotor dysfunction in the atrofish, we next investigated organization and homeostasis of the neuromuscular system. In muscle tissue, the neuromuscular junction (NMJ) mediates muscle contractions in response to neuronal stimuli. To analyze NMJ maintenance following muscle-fiber degeneration in the atrofish, we first performed axonal and synaptic staining using anti- acetylated-tubulin (axons) and SV2 and znp-1 (pre-synapses) antibodies. We observed reduced peripheral nervous system innervation and synaptic density in muscle tissue at 6 dpf after 24 or 48 hours of 4-OHT treatment compared to control larvae (Fig 2C and S4A in S4 Fig). Interestingly, we found that reduced muscle-tissue innervation correlates positively with muscle-fiber degeneration (Fig 2C and S4A in S4 Fig), suggesting that disrupting synapse stability could also potentially promote motoneuron degeneration. To test this hypothesis, we analyzed motoneurons maintenance using the *Tg(mnx1:mCherry)* line and revealed that following muscle-fiber degeneration in atrofish larvae, NMJ density is reduced in the atrophic area and this phenotype is associated with reduced motoneuron-cell-body density in the spinal cord (Fig 2C-2D). We observed motoneuron degeneration in the spinal cord in 19%, 11%, and 8% of the atrofish larvae after 72, 48, and 24 hours of 4-OHT treatment, respectively (S4B in S4 Fig). Using the *Tg(mnx1:mCherry)* line, we analyzed the neuronal degenerative phenotype of atrofish larvae by quantifying the global volume of motoneuron cell bodies in the spinal cord and axons in

the peripheral system, axon branching, and NMJ density in control (wild-type larvae treated with DMSO or 4-OHT, and atrofish larvae treated with DMSO) and 4-OHT-treated atrofish larvae (Fig 3A-A" and S1 and S2 Movies). Finally, using HCR RNA-FISH for the *mpeg1.2* gene on the *mnx1:mCherry* genetic background labeling motoneurons, we identified macrophages in the degenerating muscle tissue, particularly in close vicinity to fragmented motoneuron axons and NMJ, with some macrophages showing GFP and mCherry fluorescence (Fig 3B). These HCR RNA-FISH results suggest that, based on *mpeg1.1*:mCherry expression, immune-cell numbers do not differ significantly between control and atrofish larvae, but that the localization of *mpeg1.2+*macrophages may differ between control and atrofish larvae. These results in turn suggest that immune cells are removing degenerated muscles and neurons, and that inflammation is associated with neuromuscular degeneration. Surprisingly, we also determined that muscle-fiber degeneration is a prerequisite to degeneration not only of motoneurons but also of other spinal neurons such as sensory neurons and interneurons, as revealed by the absence of HuC/D expression (a generic marker of postmitotic neurons) and SV2 (pre-synapses) labelling in the spinal cord (S4C in S4 Fig). Concordantly, we have observed no examples of neuron degeneration in the presence of muscle fibers (the latter revealed by phalloidin or A4.1025 staining), and we showed that spinal-neuron degeneration correlates with the absence of muscle fibers (S4D in S4 Fig). Finally, we demonstrated that dorsal root ganglia in the peripheral sensory system or neurons in the gut are not affected by muscle-fiber degeneration (S4D in S4 Fig), suggesting that the neuronal cell bodies degenerative phenotype is specific to the spinal cord in the central nervous system. Altogether, these data indicate that the muscle-fiber degeneration underlying muscle atrophy can lead to neurodegeneration in both the peripheral (NMJ) and central nervous system (cell bodies).

While it has been postulated, tested, and widely accepted that denervation and loss of functional synaptic interactions at the NMJ has an impact on muscle-fiber homeostasis with the potential to lead to muscle atrophy [41–44], the idea that the reverse sequence of events, i.e., muscle atrophy could lead to neurodegeneration, has not been thoroughly investigated. To our knowledge there is currently no direct evidence that this "reverse" process takes place. Our experiments above, conducted in a zebrafish model of muscle degeneration, reveal that muscle-specific expression of Atrogin-1 leading to skeletal-muscle atrophy can indirectly negatively impact the central nervous systems and spinal-neuron survival. This observation suggests a new pathological component in the etiology of neuromuscular degeneration, and potentially sarcopenia, and raises the intriguing possibility of a detrimental and degenerative feedback loop between muscle fibers and motoneurons during muscle aging.

## Muscle dysfunction is reversible in atrofish larvae

The above analysis demonstrates that Atrogin-1 expression in zebrafish larvae underlies neuromuscular degeneration and locomotion dysfunction. We uncovered some of the molecular and cellular mechanisms associated with these defects. We demonstrated that myosin light-chain degradation, muscle atrophy, and neurodegeneration are associated with increased proteolysis, cellular stress, inflammation, and ECM remodeling following Atrogin-1 expression in muscle fibers. To determine whether or not these phenotypes are reversible, we analyzed the recovery of locomotion capacities in atrofish larvae following discontinuation of 4-OHT treatment. We first demonstrated that 4-OHT-treated atrofish larvae show reduced locomotion between 6 and 8 dpf compared to DMSO-treated atrofish larvae and 4-OHT-treated wild-type (WT) larvae (Fig 4A and S3 Movie). Treatment with 4-OHT has no significant impact on cardiovascular-system function in WT or atrofish larvae, as cardiac volume, heartbeat, and blood flow are comparable between the two conditions (Fig 4C and 4D, and S4 and S5 Movies), suggesting that neuromuscular dysfunction is the main driver of reduced locomotion. We next treated atrofish larvae with DMSO or 4-OHT for 24 hours (between 5 and 6 dpf, larvae were treated with DMSO or 4-OHT for 24 hours before the locomotion test), and then placed all larvae in chemical-free medium for a 48-hour locomotion test. We observed that following placement in chemical-free medium, the 4-OHT-treated atrofish larvae recovered locomotion functions and performed similarly to the DMSO-treated atrofish larvae (Fig 4B and S6 Movie). Interestingly, our results also showed that after recovery, 4-OHT-treated atrofish larvae exhibited a circadian pattern of activity identical to that

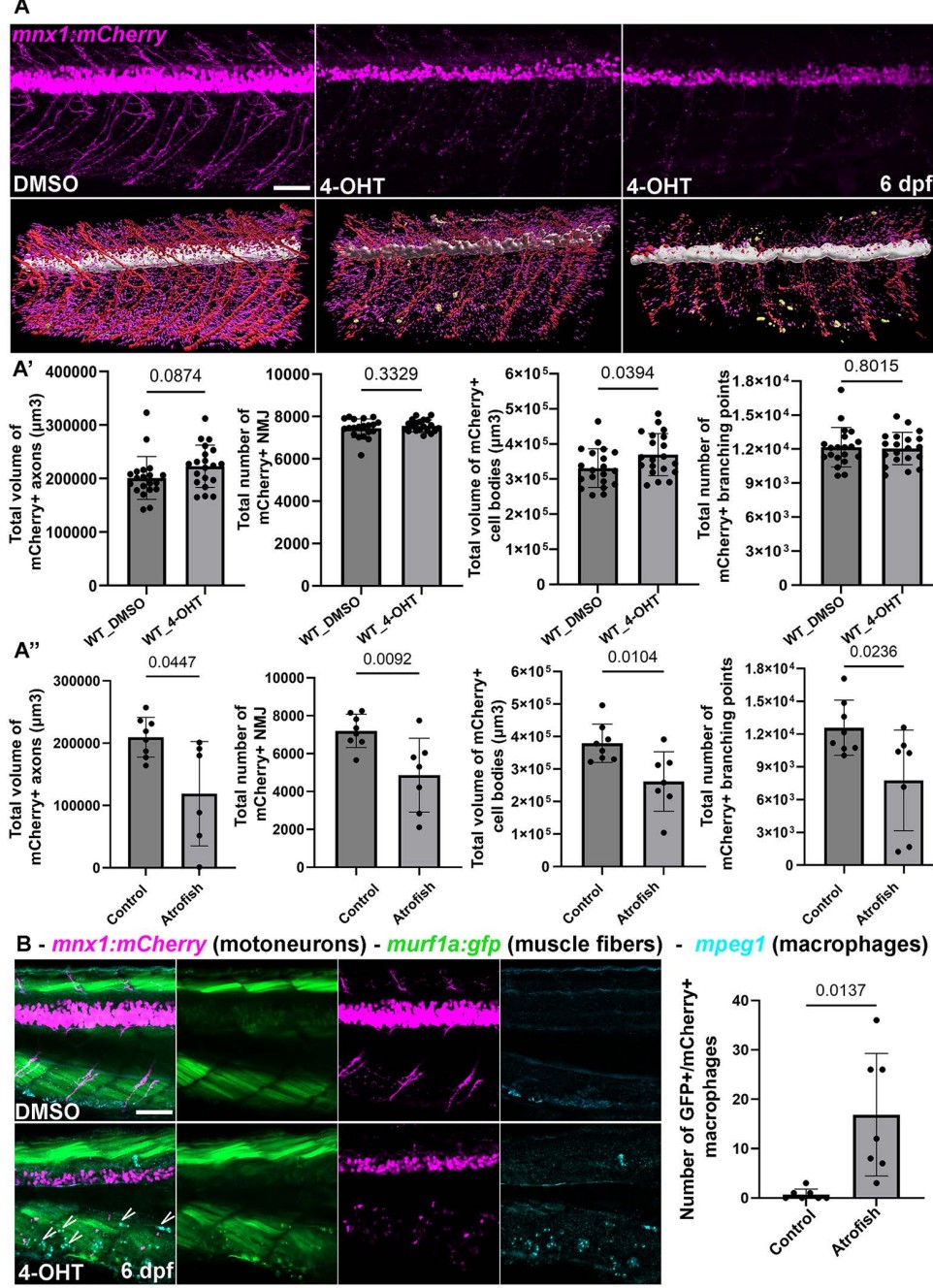

**Fig 3. Muscle-specific Atrogin-1 expression affects motoneurons in the central and peripheral nervous system. (A)** Confocal projections (top) and 3D reconstructions (bottom) of control or atrofish larvae treated with 4-OHT for 48 hours at 6 dpf. (A', A") Analysis of the motoneuron system reveals that motoneurons, axon branching, and neuromuscular connectivity and density (NMJ) are affected in atrofish (DMSO; n = 8 and 4-OHT; n = 7) but not in WT larvae treated with 4-OHT (DMSO; n = 20 and 4-OHT; n = 20). Each biological component was isolated and assigned a unique color label: white (spinal cord), red (axons), magenta (NMJs) and yellow (immune cells *mpeg1.2*+macrophages in Fig 3B). Statistical significance was determined by t-test, two-tailed, unpaired with Welch's correction. **(B)** Confocal sections of DMSO-treated atrofish larvae (top) or atrofish larvae treated with 4-OHT (bottom) for 48 hours at 6 dpf showing macrophages in close vicinity of the motoneuron axons and NMJ (*mnx1:mCherry*) in the degenerating muscle tissue (*murf1a:GFP*). Arrows show *mpeg1*+macrophages expressing both GFP and mCherry (DMSO; n = 7 and 4-OHT; n = 7). Statistical significance is determined by t-test, two-tailed, unpaired with Welch's correction. Error bars represent s.d. Scale bars: 100 μm.

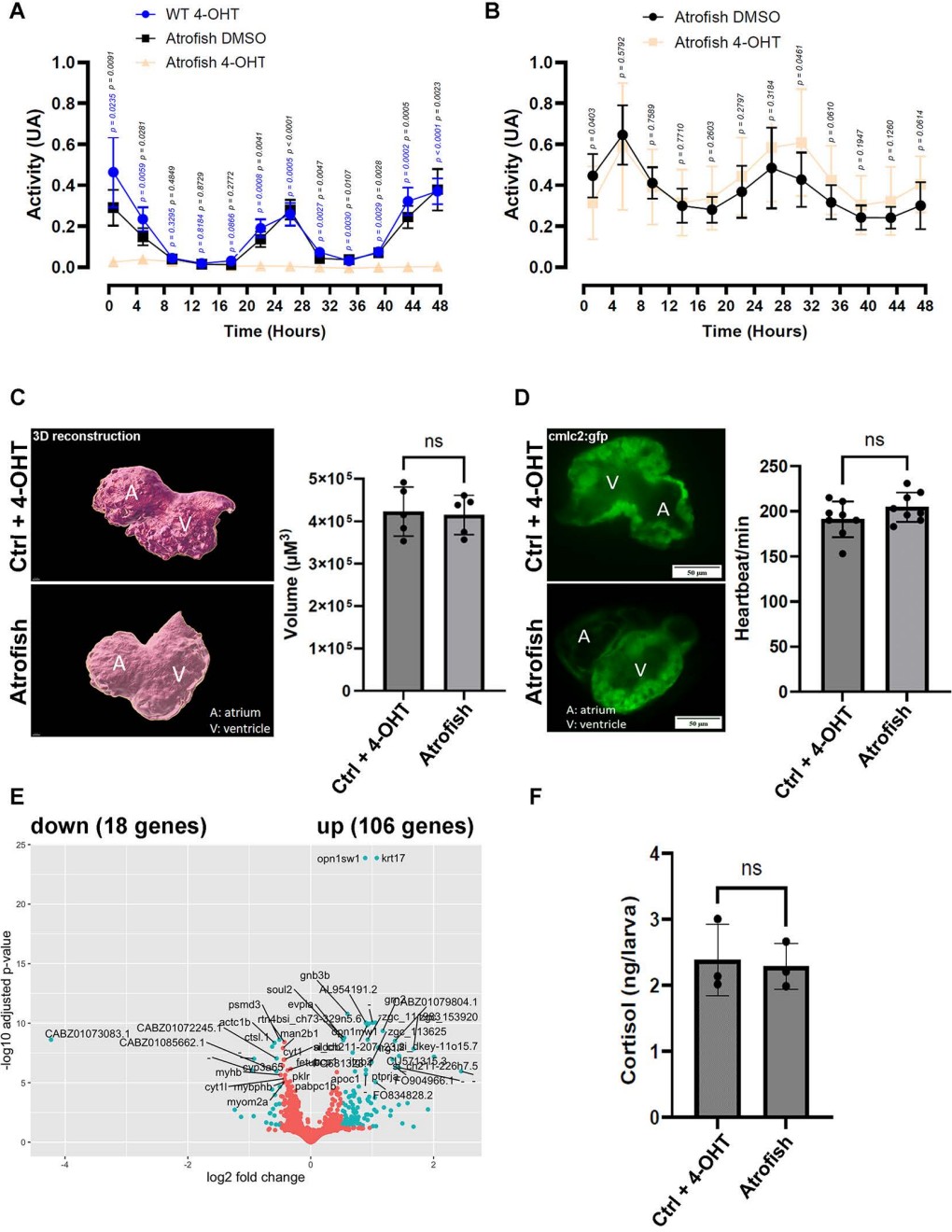

**Fig 4. Atrogin-1-dependent muscle dysfunction is reversible in atrofish larvae. (A, B)** Quantification of locomotion activity in zebrafish larvae. **(A)** WT larvae treated with 4-OHT (n = 8) and atrofish larvae treated with DMSO (n = 8) or 4-OHT (n = 8). Larvae were treated with DMSO or 4-OHT for 24 hours between 5 and 6 dpf before the locomotion test, and for 48 hours during the test. The figure shows only the 48-hour test results. **(B)** Atrofish larvae treated with DMSO or 4-OHT. Larvae were treated with DMSO (n = 12) or 4-OHT (n = 11) for 24 hours between 5 and 6 dpf before the locomotion test, and then placed in chemical-free medium for 48 hours during the locomotion test. The figure shows only the 48-hour test results. Statistical significance is determined by multiple t-test comparison with False Discovery Rate and individual variance for each time point, two-tailed, unpaired. **(C, D)** Quantification of heart volume after 3D reconstruction in control fish (n = 5) or atrofish (n = 5) **(C)** or heartbeat using the *cmlc2:gfp* transgene expressed in cardiomyocytes **(D)** at 6 dpf in control fish (WT fish) (n = 8) or atrofish (n = 8) treated with 4-OHT for 24 hours between 5 and 6 dpf. Statistical significance is determined by multiple t-test, two-tailed, unpaired. **(E)** Graphical representation of bulk RNAseq analysis and variation in gene-expression profiles in DMSO-treated control larvae (n = 4; 50 larvae/sample) vs. atrofish (n = 4; 50 larvae/sample) after 24 hours of 4-OHT treatment between 5 and 6 dpf, followed by 2 days of recovery until 8 dpf. **(F)** Quantification of cortisol expression levels in control larvae (4-OHT-treated; n = 3 with 30 larvae/sample

n = 30) and 4-OHT-treated atrofish larvae (n = 3 with 30 larvae/sample) after recovery. Statistical significance is determined by t-test, two-tailed, unpaired. Error bars represent s.d. Scale bars: 50 μm.

of DMSO-treated atrofish larvae, suggesting that that their circadian clock is functional. Together, these results indicate that the locomotion-defect phenotype is reversible and suggests that atrofish larvae are able to recover from their muscle and neural deficits. While the question of whether the rapid swimming recovery is driven by a regenerative response still requires investigation, the timing suggests that preventing *atrogin-1* overexpression in muscle fibers limits the death of additional muscle fibers and allows for an efficient return to a homeostatic and functional state. Indeed, while most, if not all, larvae exhibit defective locomotion, only a small percentage of atrofish exhibit severe neuronal degeneration and/or muscle degeneration after 24 hours of 4-OHT treatment (S1B and S4B in S1 and S4 Figs). Together, these observations indicate that mechanisms other than *atrogin-1* overexpression could underlie the defect in locomotion. Because our transcriptomic analysis identified inflammation, the stress response, and ECM remodeling as biological processes significantly dysregulated in atrofish, we hypothesized that these cellular mechanisms could also participate in the reduced swimming capacity phenotype. In addition, if these cellular processes do indeed contribute to defective swimming in atrofish, they could potentially be reversed more quickly than would be possible via the level of cell regeneration required to enable locomotion recovery. We therefore hypothesize that these cellular mechanisms could underlie recovery of swimming function. Further experiments using anti-inflammatory molecules or fish mutants with knockout of the glucocorticoid receptor gene would allow testing of these hypotheses.

To characterize the recovery response at the molecular level, we performed gene-expression analysis at 8 dpf using mRNA from atrofish larvae treated with either DMSO or 4-OHT between 5 and 6 dpf and then placed in chemical-free medium for 48 hours. We observed that only 124 genes are significantly dysregulated at 8 dpf, i.e., after 48 hours of recovery (Fig 4E and S2 Table), suggesting that 4-OHT-treated atrofish larvae progressively returned to homeostatic conditions following their return to chemical-free medium between 6 and 8 dpf. We then performed GO and KEGG analyses using our gene-expression results. Interestingly, immune and defense responses remain highly represented in 4-OHT-treated atrofish larvae after 48 hours of recovery (S5A and S5B in S5 Fig), but markers for macrophage and neutrophil populations, *mpeg* and *mpx*, respectively, are no longer significantly upregulated at this time (S2 Table). In addition, some pro-inflammatory genes that are significantly upregulated at 6 dpf, including *ccl34*, *cxcl8*, *cxcl18*, *mmp9*, and *il1b*, are also no longer significantly upregulated at 8 dpf. These data suggest that shortly after recovery of 4-OHT-treated atrofish larvae from locomotion defects, the pro-inflammatory response is reduced in favor of a pro-regenerative (or anti-inflammatory) response, as indicated by high expression of the IL-22 receptor gene (IL-10 family receptors [45]). Stress-response genes, glucocorticoid-receptor-target genes and cortisol are also no longer overexpressed at 8 dpf in atrofish larvae (S2 Table and Fig 4F). Together, the above results indicate that the recovery of locomotion capacity in 4-OHT-treated atrofish larvae correlates with reduced stress and pro-inflammatory responses.

### Young adult atrofish exhibit reduced muscle mass and swimming performance

To determine whether the atrophic phenotype that we observe in zebrafish larvae and that is dependent on Atrogin-1 expression in skeletal muscles is recapitulated in adult fish, we treated one-month-old fish carrying Tg(*ubi:Lox-stop-Lox-atrogin1*) and Tg(*503unc:creERT2*) transgenes with 1.25μM of tamoxifen (TAM), once every week until 6 months of age. To analyze muscle atrophy, we performed tissue-clearing of whole adult muscle tissue [46]. In combination with MesoSPIM light-sheet microscopy for macrosamples [47], tissue-clearing allowed us to assess and quantify muscle integrity by imaging the fluorescence of the muscle tissue by GFP expression analysis in muscle fibers from the *murf1a:gfp* transgene. We observed a significant number of areas in the muscle tissue of TAM-treated atrofish with muscle-fiber degeneration (Fig 5A). Volumetric analysis using 3D reconstruction reveals that the volume of GFP+ muscle tissue in the atrofish is ~37% less

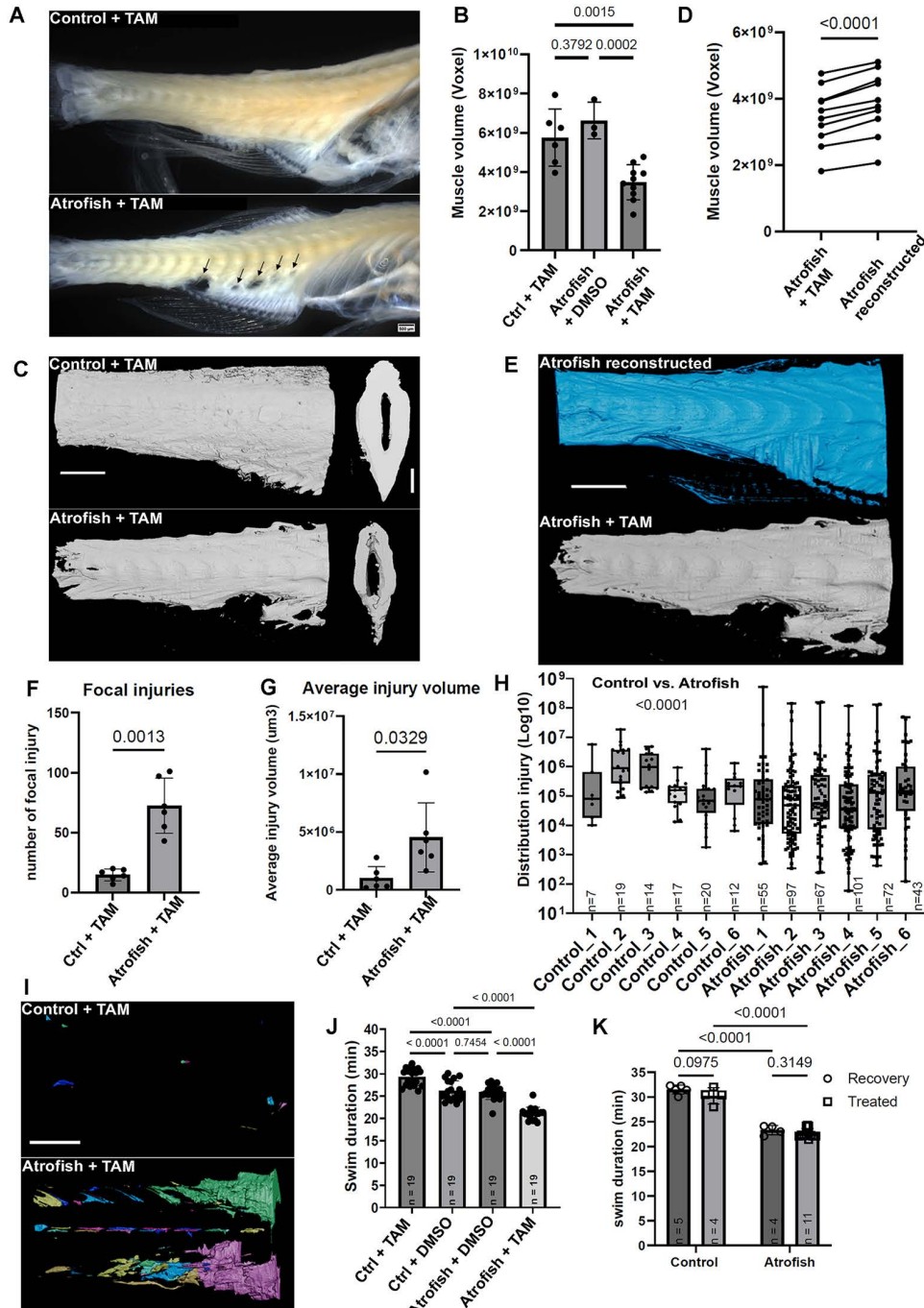

**Fig 5. Atrogin-1-dependent muscle dysfunctions are irreversible in chronically treated adult atrofish. (A)** Brightfield imaging of adult zebrafish trunk after tissue-clearing showing degeneration of skeletal-muscle tissue in one-month-old atrofish treated with tamoxifen (TAM) for 5 months (bottom) compared to TAM-treated control fish (control; top) at 6 months of age. Arrows indicate regions of degeneration. **(B)** Quantification of the volume of muscle tissue in TAM-treated control fish (n = 6) or atrofish treated with either DMSO (n = 3) or TAM (n = 10). Statistical significance is determined by t-test, two-tailed, unpaired. **(C)** Representative images of 3D reconstructed trunk skeletal-muscle tissue after chronic TAM treatment in control fish (top) or atrofish (bottom) at 6 months of age. **(D)** Quantification of the volume expected after 3D reconstruction of atrofish trunk skeletal-muscle tissue (n = 10). Statistical significance is determined by t-test, two-tailed, paired. **(E)** Representative images of 3D reconstructed trunk skeletal-muscle tissue after chronic TAM treatment in atrofish at 6 months of age. **(F, G)** Quantification of the number of focal injuries in TAM-treated control fish (n = 6) and TAM-treated atrofish (n = 6) at 6 months of age after 3D construction **(F)**. Quantification of the average volume of injuries in TAM-treated control fish (n = 6) and

TAM-treated atrofish (n = 6) at 6 months of age after 3D construction **(G)**. Statistical significance is determined by t-test, two-tailed, unpaired with Welch's correction. **(H)** Quantification of the distribution of the injuries identified in **(G)** in TAM-treated control fish (n = 6) and TAM-treated atrofish (n = 6) at 6 months of age after 3D construction. n in the graphic indicates the number of injuries per biological replicate. Injuries from 6 biological replicates for control or atrofish have been pulled together to determine statistical significance using t-test, two-tailed, unpaired. **(I)** Representative images of the numbers and sizes of injuries in trunk skeletal-muscle tissue after chronic TAM treatment in control (top) or atrofish (bottom) at 6 months of age. Each color represents a single injury, quantified in **(H)**. **(J)** Quantification of swimming capacity of control fish and atrofish at 6 months of age treated with either DMSO or 4-OHT for 5 months (n = 19 for each condition). Statistical significance is determined by Mann-Whitney test, two-tailed, unpaired. **(K)** Quantification of swimming capacity of control fish (n = 9) and atrofish (n = 15) at 9 months of age, treated with TAM for 5 months, i.e., between 1 and 6 months of age, and then treated with either DMSO (ctrl, n = 5; atro, n = 4) or TAM (ctrl, n = 4; atro, n = 11) for an additional 3 months (i.e., until 9 months of age), to test for recovery of swimming function. Statistical significance is determined by a Two-way Anova. Error bars represent s.d. Scale bars: 1 mm.

than that of control animals (Fig 5B and 5C). This observation indicates that chronic Atrogin-1 expression in young zebrafish leads to early muscle atrophy and degeneration. In addition to the reduction in muscle volume, we also quantified a 13% reduction in muscle tissue expressing GFP in the atrofish compared to the expected calculated volume after 3D reconstruction (Fig 5D and 5E), suggesting that the atrofish exhibit not only an atrophic phenotype but also reduced muscle mass and muscle degeneration inside the tissue. We identified an average of ~75 sites of injury in the atrofish versus ~10 in control fish (Fig 5F), with an average muscle-degeneration volume of ~$4.11 \times 10^6$ $\mu m^3$ in atrofish versus ~$1.06 \times 10^6$ $\mu m^3$ in control fish (Fig 5G). The distribution of the injury sites indicates that atrofish harbor larger numbers of both small-volume and large-volume injuries compared to control fish at 6 months of age (Fig 5H and 5I). We then calculated the percentages of injuries at different positions in the truck (at each position, the percentage of the total number of injuries in the trunk), and the average injury volume at each position, to correlate the sites of degeneration with the position of the fins. Interestingly, we observed that adult atrofish have ~50x, ~15x, and ~3x greater amounts of degenerated tissue than control fish at the anal, dorsal, and caudal fins, respectively (S5C in S5 Fig). The above observation suggests that muscle contraction is involved in the development of the degenerative phenotype, with greater numbers of larger injuries at the anal and dorsal fins in the adult atrofish. In addition, we observed a slight but significant reduction in the size of TAM-treated atrofish compared to TAM-treated control fish, 6 months after treatment (S5D and S5E in S5 Fig). To determine if muscle atrophy and degeneration are associated with reduced muscle function, we tested swimming performance. We showed that at 6 months of age, swimming performance of atrofish is reduced by 31% compared to that of control fish (Fig 5J), indicating that muscle atrophy correlates with reduced muscle function in young animals (males and females perform similarly, for both control fish and atrofish; S5F in S5 Fig). However, we observed that control fish treated with TAM until 6 months of age show a slight, but significant, increase in swimming performance compared to control fish treated with DMSO until 6 months of age (Fig 5J). This effect could be attributed to a moderate positive impact of TAM treatment on muscle tissue, as studies in mice have suggested that TAM could have a beneficial effect on muscle hypertrophy and function [48–51]. We also demonstrated that 4-OHT treatment of WT larvae has no negative impact on neuronal survival or neuromuscular-system organization. Rather, we observed a slight increase in motoneurons after 4-OHT treatment of WT larvae. Together, these observations suggest that neither TAM nor 4-OHT treatment has a significant negative effect on the zebrafish neuromuscular system that would be responsible for locomotion defects. Next, we asked if, as we observed in larvae, the locomotor phenotype is reversible in adult fish. To test this, we allowed some fish previously treated with TAM between 1 and 6 months of age to recover by maintaining the fish without treatment between 6 and 9 months of age. Importantly, both males and females again showed similar swimming performances in each experimental condition (S5G in S5 Fig). Interestingly, we observed that the swimming performance of "recovered" nine-month-old atrofish is still poorer than that of control fish (Fig 5K). These data indicate that 3 months after stopping the TAM treatment, an efficient regenerative response does not take place in adult atrofish. This observation raises the hypothesis that the muscle stem-cell pool is dysfunctional (senescence?) or absent (exhaustion?) in these animals, potentially because of continuous muscle-fiber degeneration associated with chronic inflammation and stress. Future studies will explore the mechanisms underlying the non-regenerative phenotype in the adult atrofish after chronic Atrogin-1 expression.

Together, our work validates the atrofish as a new genetic model of muscle atrophy, both in zebrafish larvae and adults (Fig 6). We demonstrated that muscle-specific expression of Atrogin-1 leads to dysregulation of processes associated with proteolysis, inflammation, the stress response, and apoptosis. We also identified myosin light-chain degradation as a

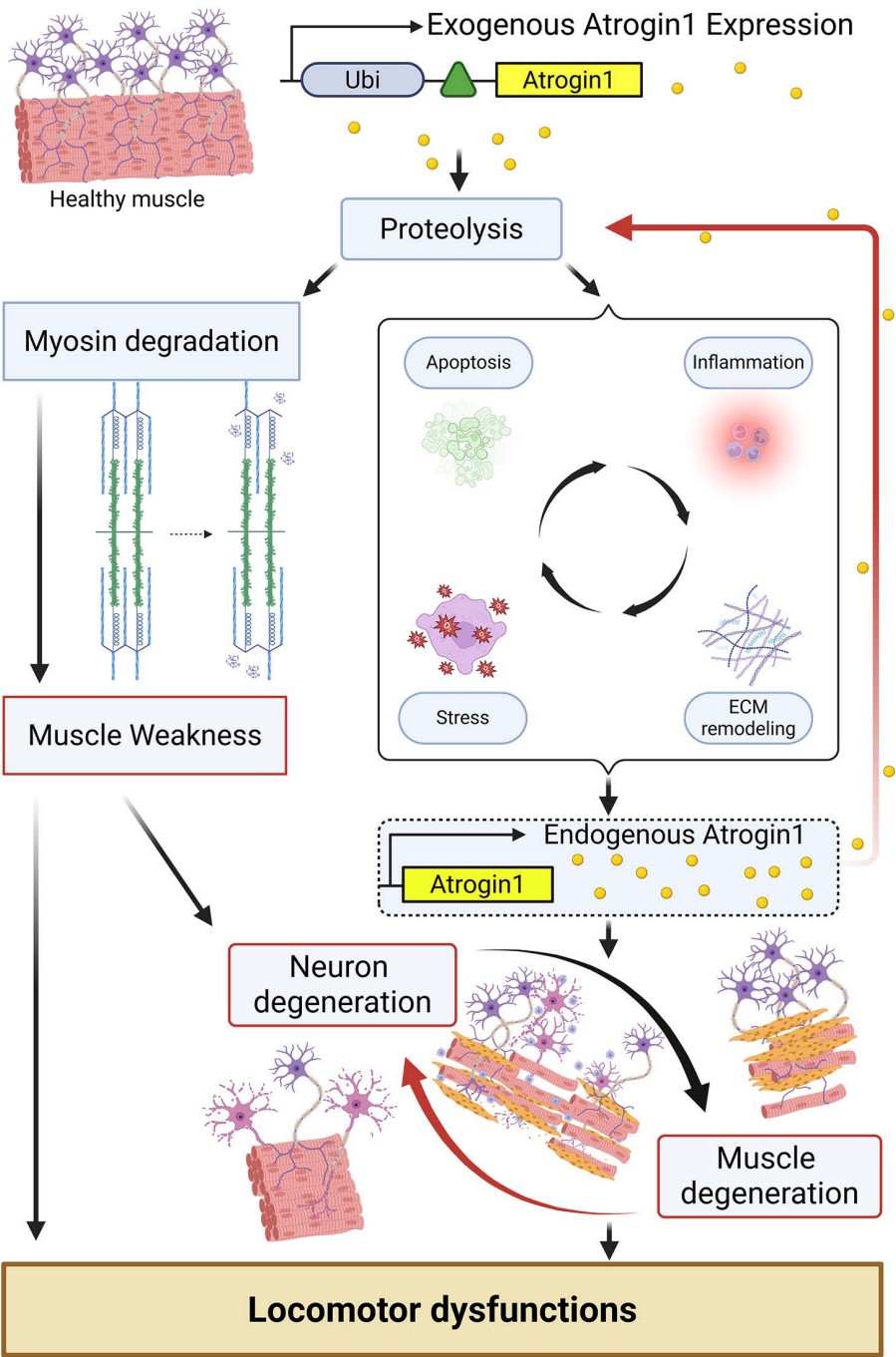

**Fig 6. Graphical representation of the biological processes underlying the neuromuscular degeneration phenotype associated with Atrogin-1 expression.** Created in BioRender. Menard, R. (2025) - https://BioRender.com/1kpf9fg.

potential driver of muscle weakness and reduced muscle function. Our observations that many injury sites are located at the anal and/or dorsal fins in both larvae and adult atrofish also suggest that muscle contraction and locomotion behavior are involved in development of the degenerative phenotype. Unexpectedly, muscle-fiber degeneration was correlated not only with NMJ reduction but also with spinal-neuron degeneration. The question of whether or not this phenotype also occurs during degenerative diseases such as sarcopenia in mammals remains unanswered.

In conclusion, the atrofish is a new genetic model of accelerated neuromuscular degeneration. Determination of its value as a proper model of accelerated muscle aging will require further investigation. Currently, the atrofish provides opportunities to better understand the biological mechanisms underlying degenerative phenotypes associated with Atrogin-1 expression as observed during sarcopenia development in mammals. It also represents a vertebrate genetic model, in larvae and young adult fish, for testing and screening drugs for their therapeutic potential to limit neuromuscular degeneration and/or promote regeneration and alleviate some of the symptoms associated with muscle atrophy and aging.

## Materials and methods

### Ethics statement

Experiments were performed in accordance with animal care guidelines. Our IACUC animal protocol (AUP#22–06) is approved by the MDI Biological Laboratory IACUC ethics committee (chair: Wynne Guglielmo; veterinarian: Peter Merril).

### Study design and statistical analysis

Research subjects (larval stage, adults vs. larvae, and genotype) and treatments applied are defined in the figures and/or figure legends. Experimental analyses were performed with a minimum of 3 biological replicates (and a minimum of 3 technical replicates when applicable). Regarding statistical power, we calculated the effect size with Cohen's D and found that all measured differences had D-values that exceeded an absolute value of 4, which qualifies as a very large effect. For experiments using zebrafish larvae as one biological replicate, a minimum of 20 larvae were used. The number of biological replicates and larvae used for each specific experiment as well as the statistical test used to calculate significance is indicated in the figure legends. For statistical analysis, p-values are indicated in the figures. Comparisons were conducted to determine significant differences between control and experimental conditions. Statistical analyses were performed using GraphPad Prism 10. The normality of data distribution was assessed using the Shapiro-Wilk test and visualized with quantile-quantile (Q-Q) plots. Homogeneity of variance between groups was evaluated using F-tests. Based on these preliminary analyses, Student's T-test, T-test with Welch's correction, or Mann-Whitney test were selected for subsequent statistical comparisons, as indicated in the figure legends. Statistical significance was set at $p < 0.05$. Processing and method analysis of the bulk RNAseq dataset is described below in the Materials and Methods section.

### Fish lines and developmental conditions

The following transgenic lines have been used in this study: Tg(*mpeg1.1:mCherry*) [52], Tg(*mnx1:mCherry*) [53], and Tg(*gfap:gfp*) [54], Tg(*ubi:Lox-stop-Lox-atrogin1*), Tg(*503unc:creERT2*) and Tg(*murf1a:gfp*). Embryos were raised and staged according to standard protocols [51]. For imaging analysis, larvae were fixed overnight at 4°C in 4% paraformaldehyde/1xPBS, after which they were dehydrated through an ethanol series and stored at −20°C until use.

### Plasmid construction and transgenic line establishment

For generation of Tg(*ubi:Lox-stop-Lox-atrogin1*) zebrafish, a p5E entry vector containing the ubiquitin promoter and the *lox*P cassette [24] was used. The *atrogin-1* ORF was amplified by PCR from zebrafish cDNA and cloned in the pME middle-entry vector. Generation of the Tg(*503unc:creERT2*) was perfomed using the small 503unc skeletal-muscle-specific promoter [23] and the creERT2 coding sequence [25].

For generation of Tg(*murf1a:gfp*) zebrafish, we cloned 5kb of the *murf1a* gene promoter in the p5E entry clone and used the pME-*gfp* plasmid as middle-entry vector.

The appropriate entry and middle-entry clones were mixed with the SV40pA 3' entry vector and recombined into the Tol2 transposon destination vector. To establish stable transgenic lines, plasmids were injected into one-cell-stage embryos with Tol2 transposase mRNA [22].

## Pharmacological treatments

Stock solution of 1-phenyl 2-thiourea (PTU; Sigma) was prepared at 25X (0.075%). PTU 1X working solution was used to inhibit pigmentation in zebrafish embryos. A stock solution of hydroxytamoxifen (4-OHT; Sigma) was prepared at 10 mM in 100% DMSO (Sigma). For induction of Atrogin-1 expression, zebrafish larvae were treated at a concentration of 10 µM.

## Gene-expression analysis

HCR-RNA fish were performed using *atrogin-1*, *pdgfrb* or *mpeg1.2* probes designed by Molecular Instruments, and *in situ* hybridization and revelation were performed following the provider protocol.

Total RNA samples for bulk RNAseq analysis were prepared with ~50 larvae/sample. Each experimental condition was performed using biological quadruplicates to ensure reproducibility. Trizol (Invitrogen) was used to homogenize cellular extracts with the TissueLyzer (Qiagen). RNA purification was performed as described above. For bulk RNAseq analysis, library preparation and sequencing were carried out by NovoGene using the Illumina NGS platform. Raw FASTQ sequencing data underwent preprocessing to filter out adapters, poly-N tails, and low-quality reads. Indexing and alignment were performed using HISAT2 (v2.0.5) [55] against the GRCz11 (*Danio rerio*) reference genome obtained from Ensembl. Mapped reads were assembled using StringTie (v1.3.3b) [56], and quantification was performed using FeatureCounts (v1.5.0-p3) [57]. Differential expression analysis was conducted in the R environment using DeSeq2 [58], and Gene Set Enrichment Analysis (GSEA) was performed using clusterProfiler [59]. All visualizations were generated using ggplot2. RNAseq data are available via the GEO database and the accession number GSE274353.

To map the effects of cortisol exposure to cell and tissue type we used the procedure reported previously [60]. Briefly, we used a published single-cell atlas of zebrafish development [61], reducing the atlas to the data from 5 dpf embryos. The 5dpf data was reclustered and marker genes identified, using Seurat v5 [62,63]. Probable cell types within the clusters were identified by matching the annotations from the full embryo atlas to the 5 dpf single-cell data. Cortisol and matched control bulk RNAseq data sets (GEO records GSE80221 and GSE144885), and quantified using the Nextflow NF-core/rnaseq pipeline (v 3.12.0 for GSE144885 data and v 3.14.0 for GSE80221 data), using the Ensembl release 110 annotations of the Zebrafish GRCz11 genome assembly. The subsequent expression matrixes were joined and converted to rlog values using DESeq2 version 1.26.1 (DESeq2) [58]. The rlog matrixes were normalized on a gene-by-gene basis, by subtracting the gene-specific average and dividing by the gene-specific standard deviation. The resulting Z-transformed matrix was subjected to Principal Components analysis using SAS JMP version 18.0.1 for Mac OS. Inspection of the PCA in conjunction with metadata (experiment, treatment, replicate) indicated that Principal Component 5 captured the response to cortisol. The loading vector for this PC was reduced to only those genes that were identified as cell cluster markers as described above, and the plot of PC5 loading was also generated using JMP.

## Cortisol level analysis

Zebrafish larvae were treated with 10 µM hydroxytamoxifen from 5 to 6 dpf. At either 6 dpf or 8 dpf, samples of 30 larvae each were snap-frozen in liquid nitrogen for 3 minutes and stored at -80°C until analysis. Cortisol levels were measured using the Cortisol ELISA kit (Neogen - 402710). For sample preparation, zebrafish larvae were lysed, homogenized, and resuspended in 250 µL extraction buffer. 200 µL of the homogenate was transferred to a glass tube and mixed with 1 mL of Ethyl Acetate by vortexing for 30 seconds. After phase separation, the upper phase was transferred to a new glass tube

and evaporated under nitrogen (N2) stream. The dried sample was reconstituted in 200 μL extraction buffer and assayed immediately following the procedures detailed in the Neogen Cortisol ELISA Kit manual.

### Locomotion (larvae) and swimming (adults) analysis

Larvae were treated with hydroxytamoxifen or DMSO at 10 μM from 5 dpf to 7 dpf, or from 5 to 6 dpf followed by multiple washes with fresh water to remove chemicals. Individual larvae were placed in 24- or 48-well plates containing 500 μL of medium. Locomotion assays were performed using Zebrabox and recorded using Viewpoint quantization software. Parameters used were: threshold 19, freezing 5, and burst 6, with data integration every 4 hours. A backlight (20%) and toplight (3%) were activated during the assay. Total locomotive activity per 4-hour period was calculated by summing small burst movements and comparing them with freezing activity. Maximum values did not exceed 14,400 seconds (4 hours) per integration point. Raw data were transformed into percentage of activity and analyzed using GraphPad Prism. Multiple unpaired t-tests with the False Discovery Rate (FDR) method were performed, analyzing mean and variance for each population row. Each data point represents one individual.

Swimming tunnel preparation, calibration, and maintenance were adapted from *Burris et al.*[64].The swim performance assay was modified to use velocity increments of 5 cm/s every 3 minutes for a total duration of 60min, with fish size positioned at 3 cm in length and an average of 5 zebrafish swimming simultaneously in the swimming chamber. This group size was selected to avoid conflict between zebrafish while enabling accurate recording of swimming performance for each individual fish. Before starting the performance assay, zebrafish were acclimated for 15min in the swimming chamber with a velocity of 30 cm/s. Significance is determined by t-test, two-tailed, unpaired. Error bars represent s.d.

### Beating heart analysis

Samples were mounted with 1.2% low-melting agarose in a petri dish 35 mm in diameter, with a glass bottom. Images were acquired using a Spinning-disk confocal unit (CSU-W1, Yokogawa, Japan) on a Nikon inverted Ti-Eclipse microscope stand (Nikon Instruments Inc., Japan), equipped with an objective magnification 20x and numerical aperture (NA) of 0.75, immersion medium Air (dry). For live imaging experiments, the sample was placed into a Stagetop Incubator System (Okolab srl, Italy) and warmed at 28°C using the Heated Chamber for Nikon Motorized Stage (H301-NIKON-Ti-S-ER, Okolab srl, Italy). Images were collected every 15ms, using an exposure time of 10ms for a total duration of 1 minute. To count heartbeats, the Z-plot axis was used and peaks were counted manually. Significance is determined by t-test, two-tailed, unpaired. Error bars represent s.d.

### Heart volume analysis

Samples were mounted with 1.2% low-melting agarose in a petri dish 35 mm in diameter, on top of agarose drops. Images were acquired using a point scanning confocal unit (LSM 980, Carl Zeiss Microscopy, Germany) with a 20x magnification and NA of 1.00 objective lens with water immersion medium (Carl Zeiss Microscopy, Germany). For 3D reconstruction, images were 4091*4006 pixels acquired at 8-bit controlled with Zen Blue 3.1 software (Carl Zeiss Microscopy, Germany). 124 μm Z-stack images were collected with a step size of 1 μm with the Motorized Scanning Stage 130x85 PIEZO (432022-9903-000, Carl Zeiss Microscopy, Germany). Significance is determined by t-test, two-tailed, unpaired. Error bars represent s.d.

### Blood flow analysis

Samples were mounted with 1.2% low-melting agarose in a petri dish 35 mm in diameter, with a glass bottom. Images were acquired using a Spinning-disk confocal unit (CSU-W1, Yokogawa, Japan) on a Nikon inverted Ti-Eclipse microscope stand (Nikon Instruments Inc., Japan), equipped with an objective magnification 20x, numerical aperture (NA) of 0.75, immersion medium Air (dry). Samples were imaged using DIC microscopy. Images were acquired in 2048*2048

pixels, 16-bit with a Scientific CMOS Zyla 4.2 (AndorTechnology, United Kingdom) controlled with NIS AR 5.41.02 software (build 1711, Nikon Instruments Inc., Japan). Samples were placed into a Stagetop Incubator System (Okolabsrl, Italy), warmed at 28°C using the Heated Chamber, and images were collected every 500 ms for a total duration of 10 sec. Significance is determined by t-test, two-tailed, unpaired. Error bars represent s.d.

### Immunostaining

Immunostaining was performed using anti-GFP (1/1,000, Torrey Pines Biolabs), anti-DsRed (1/500, Takara), anti-HuC/D (1/500, Invitrogen), anti-znp1 (1/50, DSHB), anti-zn8 (1/50, DSHB), anti-F310 (1/50, DSHB), anti-A4.1025 (1/50, DSHB), anti-SV2 (1/50, DSHB), and anti-acetylated tubulin (1/500, Sigma) as primary antibodies and Alexa 488, Alexa 555, or Alexa 647-conjugated goat anti-rabbit immunoglobulin G (IgG) or goat anti-mouse IgG (1/1,000) as secondary antibodies (Invitrogen). Phalloidin-Alexa 488 was used to bind actin and reveal muscle fibers (Invitrogen).

### Niel red and Sirius red stainings

Niel red was used at 0.25mg/ml (0.75 mM) in 0.4% acetone for 30 min on larvae freshly euthanized with tricaine. Larvae were then washed 4 times with PBS tween 0.1% and mounted in glycerol for fluorescent imaging. Sirius red was used at 0.1% in picric acid for 60 min on fixed larvae. Larvae were then washed 4 times with 0.5% citric acid and mounted in glycerol for birefringence imaging.

### Tissue Clearing

Tissue clearing was adapted from Pende et al., [45]. Fish were euthanized in ice for 5min and then dehydrated gradually with ethanol and stored until use. Before clearing, fish were rehydrated in 0.1% PBS-Tween and meticulously skinned, and were then placed in 3% $H_2O_2$ solution for 10min to increase depigmentation efficiency. Fish were then incubated in Solution 1 (25% urea, 5% Triton-100x, 5% Chaps) for 5h at 37°C under gentle agitation. Then fish were gradually dehydrated using 50% Tert-butanol/4% PBS-Quadrol; 75% Tert-butanol/4% PBS-Quadrol, overnight at 37°C; 80% 1-propanol/4% $dH_2O$-Quadrol, overnight at 4°C; 90% 1-propanol/4% $dH_2O$-Quadrol, overnight at 4°C; and 100% 1-propanol, overnight at 4°C. Each step was performed under gentle agitation. For final RI matching fish were then immersed in sDBE consisting of DBE which was first purified with aluminiumoxde and subsequently stabilized with 0.3% propyl gallate.

### Image acquisition

Images were acquired using either a widefield Zeiss AxioImager M2 Upright microscope (430004-9902-000, Carl Zeiss Microscopy, Germany), a Zeiss Axio Observer.Z1 inverted microscope (431007-9902-000, Carl Zeiss Microscopy, Germany) or a point scanning confocal unit (LSM 980, Carl Zeiss Microscopy, Germany) on a Zeiss Axio Examiner Z1 upright microscope stand (409000-9752-000, Carl Zeiss Microscopy, Germany), as indicated in figure legends.

For birefringence analysis, we used a Polarizer D (000000-1121-813, Carl Zeiss Microscopy, Germany) and a LD condenser (424244-0000-000, Carl Zeiss Microscopy, Germany).

For light-sheet microscopy, images were acquired using a mesoSPIM light-sheet microscope [46] equipped with a MI Plan 4.0X/ 0.35 Air objective lens (Ref: 130-133-602). The images underwent modifications using Fiji software. To ensure consistency in image processing, a macro was devised in collaboration with our team and the Light Microscopy Facility at MDIBL and will be made available upon request.

### 3D reconstruction and segmentation

**Zebrafish larvae.** To analyze *atrogin-1* expression, we imaged larvae using a Zeiss LSM980 confocal microscope. Three-dimensional reconstruction was performed using Imaris software. Volumes of *atrogin-1* expression were

reconstructed and segmented using the Surfaces module in Imaris. The following parameters were applied: surface detail of 0.0529 µm with background subtraction (local contrast) using a 1.98-µm-diameter sphere. An intensity threshold of 200 and a surface filter of 100 were applied. The "Center point" surface style was selected for optimal rendering. Motoneurons were reconstructed and segmented using the Surfaces module in Imaris. Different parameters were applied for each anatomical component. Spinal cord segmentation: Surface detail of 2 µm with absolute intensity thresholding was applied. An intensity threshold of 3054, minimum volume of 5 µm³, and surface filter above 6000 were used. Axons and NMJ segmentation: Surface detail of 0.2 µm with background subtraction (local contrast) using a 1.98-µm-diameter sphere was applied. An intensity threshold of 375 and surface filter above 15 were used. Two classifications were defined at a threshold of 607 to distinguish NMJs from axons. Visualization and correction: Each anatomical component was isolated and assigned unique color labels: white (spinal cord), red (axons), magenta (NMJs) and yellow (immune cells). Auto-prediction was used to identify different motoneuron components, followed by manual segmentation correction to ensure accurate assignment. Morphometric analysis: Branching patterns were obtained using the Filaments module in Imaris with the soma, branching, and spine model. Soma were defined with an average diameter of 13.2 µm, and segments were set to range between 0.2 and 8 µm. The segment threshold was set at 100. No training data were applied. Quantitative data were extracted from the Statistics module and analyzed using GraphPad Prism. Statistical significance was determined as described in the Statistical Analysis section of the Materials and Methods and Figure legends.

**Adult zebrafish.** Fish were imaged using MesoSPIM light-sheet microscopy with both right and left lasers to increase accuracy of the final image, followed by 3D reconstruction using Amira software. Reconstruction of the 3D image required superposition and alignment of both right and left laser images. For alignment, both files were loaded and colored with two high-contrast colors (green and magenta) in the 'project' window. Three easily identifiable structures were selected, and one channel was moved along the x, y, and z axes until structures in both images matched. Following alignment, images were reoriented so that the longitudinal section of the fish could be easily visualized on an x-y axis. While optional, this latter step increased efficiency during segmentation of focal injuries in the fish trunk. Volume was reconstructed by creating a mask of GFP-positive tissue and was adjusted manually to correct for autofluorescent structures such as the spinal cord, aorta, and caudal vein. The histogram range obtained from the control reconstruction was used to reconstruct atrofish uninjured reconstructed volume, with manual adjustments as needed. For segmentation, each section was divided into interior (GFP-positive tissue) and exterior (GFP-negative) materials. Individual injuries were isolated and assigned unique color labels. Auto-selection was used for clearly defined injuries not connected to the exterior material, while manual segmentation was performed for open injuries. Videos were extracted from Amira software using Camera Orbit, Animation Director, and Volume Rendering modules and exported as.AVI files at a rate of 25 fps. Snapshots were extracted using Volume Rendering and Snapshot modules, and were exported as.PNG files. Volumes were extracted from the 'Material Statistics' module and uploaded to GraphPad Prism for statistical analysis. The significance of injury volumes and location were determined by t-test, two-tailed, unpaired. Error bars represent s.d.

## Supporting information

**S1 Fig. Proteolysis, inflammation, stress response and apoptosis processes are affected in the atrofish. (A)** Schematic representation of the method used to validate Tg(*503unc:creERT2*) transgenic line with the Tg(*ubi:switch*) transgene. Efficient recombination of the *loxP* sites following 4-OHT treatment leads to mCherry expression in muscle fibers. **(B)** Quantification of proportion of the different phenotypes, as measured via phalloidin incorporation, observed in atrofish larvae treated with DMSO (n = 29), and after 24 (n = 29), 48 (n = 28), or 72 (n = 34) hours of 4-OHT treatment. Significance is determined by contingency Fisher's exact test. **(C)** GO (Gene Ontology) analysis between control atrofish larvae (DMSO-treated) and atrofish larvae treated with 4-OHT for 24 hours. **(D)** KEGG analysis between control atrofish larvae (DMSO-treated) and atrofish larvae treated with 4-OHT for 24 hours. Error bars represent s.d. (TIF)

**S2 Fig. Inflammation, fat deposition, gliosis, and fibrosis in the atrofish.** Confocal projections of DMSO-treated atrofish larvae (top) or atrofish larvae treated with 4-OHT for 48 hours (bottom) at 6 dpf. **(A)** The number of macrophages was quantified using the *mpeg1:mCherry* transgenic line (DMSO; n = 13 and 4-OHT; n = 10). **(B)** Fat deposition was analyzed using Nile red incorporation (DMSO; n = 20 and 4-OHT; n = 20). Statistical significance is determined by multiple t-test, two-tailed, unpaired. Error bars represent s.d. **(C)** Representative picture of Sirius red staining to analyze collagen deposition in skeletal muscle tissue. **(D)** Representative images of *gfap:gfp* expression. GFAP expression was analyzed using the *gfap:gfp* transgenic line. **(E)** Representative image of *pdgfrb* expression. *pdgfrb* expression was analyzed using HCR fluorescent *in situ hybridization*. Scale bars: 100 μm.
(TIF)

**S3 Fig. Myosin light chain expression in fast-twitch muscle fibers, but not myosin heavy-chain expression, is affected in the atrofish. (A)** Apotome sections of DMSO-treated atrofish larvae (left) or atrofish larvae treated with 4-OHT for 72 hours (right) at 6 dpf showing reduced F310 immunolabelling (purple) in trunk skeletal muscles. **(B)** Apotome sections of DMSO-treated atrofish larvae (left) or atrofish larvae treated with 4-OHT for 72 hours (right) at 6 dpf showing reduced F310 immunolabelling (purple) in trunk skeletal muscles in combination with dark-field imaging revealing the shape of the skin, myotomes, and notochord. **(C)** Quantification of proportion of the different phenotypes observed in DMSO-treated atrofish larvae (n = 33) or atrofish larvae after 24 (n = 36), 48 (n = 32), or 72 (n = 28) hours of 4-OHT treatment, with phalloidin incorporation and F310 immunostaining. Significance is determined by contingency Fisher's exact test exact. **(D)** Confocal sections of DMSO-treated atrofish larvae (top) or atrofish larvae treated with 4-OHT for 24 hours (bottom) at 6 dpf showing phalloidin incorporation (green), A4.1025 (purple) and F310 (cyan) immunolabelling in trunk skeletal muscles. White arrow shows phalloidin incorporation in the absence of F310 expression but presence of A4.1025 expression. Asterisks show sites of muscle-fiber degeneration. Scale bars: 100 μm.
(TIF)

**S4 Fig. Survival of spinal neurons, but not of dorsal root ganglia, is affected in the atrofish. (A)** Apotome sections of DMSO-treated atrofish larvae (left) or atrofish larvae treated with 4-OHT for 48 hours (right) at 6 dpf showing reduced acetylated tubulin (purple) and A4.1025 (green) immunolabelling in trunk skeletal muscles. Asterisks show sites of muscle-fiber degeneration. **(B)** Quantification of proportion of the different phenotypes observed in DMSO-treated atrofish larvae (n = 32) or atrofish larvae after 24 (n = 38), 48 (n = 35), or 72 (n = 32) hours of 4-OHT treatment, with HuC/D immunostaining showing spinal neurons. Significance is determined by contingency Fisher's exact test exact. **(C)** Confocal sections of DMSO-treated atrofish larvae (left) or atrofish larvae treated with 4-OHT for 24 hours (right) at 6 dpf showing phalloidin incorporation (green), HuC/D (purple), and SV2 (cyan) immunolabelling in trunk skeletal muscles. Bottom row shows phalloidin, HuC/D, and SV2 staining in the central region of the trunk muscles shown in the top row. **(D)** Apotome sections of DMSO-treated atrofish larvae (left) or atrofish larvae treated with 4-OHT for 72 hours (middle and right) at 6 dpf showing reduced HuC/D (purple) and A4.1025 (green) immunolabelling in trunk skeletal muscles. Asterisks show sites of muscle-fiber and spinal-neuron degeneration. Dorsal root ganglia in the peripheral nervous system (top arrow) and neurons in the gut (bottom arrow) are not affected by muscle degeneration. Scale bars: 100 μm.
(TIF)

**S5 Fig. Atrofish larvae and adults differ in their capacity for recovery. (A)** GO (Gene Ontology) between atrofish larvae treated with DMSO for 24 hours and atrofish larvae treated with 4-OHT for 24 hours, with all larvae then placed in chemical-free medium for 48 hours until 8 dpf. **(B)** KEGG analysis between atrofish larvae treated with DMSO for 24 hours and atrofish larvae treated with 4-OHT for 24 hours, with all larvae then placed in chemical-free medium for 48 hours until 8 dpf. **(C)** Schematic representation of injury locations and distribution (distribution = number of injuries in a

particular location as a percentage of the total number of injuries) in control fish (black) and atrofish (red) treated with TAM for 5 months until 6 months of age: ratio of injury size (top), number of injuries (middle), and volume of injuries (bottom) between control and atrofish. The red line separates the different muscle regions of interest for the analysis. **(D)** Quantification of fish weight in control fish (n = 19) treated with tamoxifen (TAM) or atrofish treated with either DMSO (n = 21) or TAM (n = 21). Statistical significance is determined by Mann-Whitney test, two-tailed, unpaired. **(E)** Quantification of fish length in TAM-treated control fish (n = 19) or atrofish treated with either DMSO (n = 21) or TAM (n = 21). Statistical significance is determined by Mann-Whitney test, two-tailed, unpaired. **(F)** Quantification of swimming capacity of control fish and atrofish at 6 months of age, treated with either DMSO (n = 9) or 4-OHT (n = 15) for 5 months, represented by gender. Statistical significance is determined by a Two-way Anova. **(G)** Quantification of swimming capacity of control fish and atrofish at 9 months of age, treated with TAM for 5 months until 6 months of age, and then treated for with either DMSO (n = 9) or TAM (n = 15) for an additional 3 months (until 9 months of age) to test for recovery of swimming function, represented by gender. Statistical significance is determined by a Two-way Anova. Error bars represent s.d. S5C in S5 Fig: Created in BioRender. Menard, R. (2025) - https://BioRender.com/asky5az.
(TIF)

**S1 Table. List of genes dysregulated in the atrofish at 6 dpf after 4-OHT treatment between 5 and 6 dpf.**
(XLSX)

**S2 Table. List of genes dysregulated in the atrofish at 8 dpf after 4-OHT treatment between 5 and 6 dpf, and recovery between 6 and 5 dpf.**
(XLSX)

**S1 Movie. Representative 3D reconstruction of the neuromuscular system of control larvae at 6 dpf after 4-OHT treatment between 5 and 6 dpf.**
(MP4)

**S2 Movie. Representative 3D reconstruction of the neuromuscular system of atrofish larvae at 6 dpf after 4-OHT treatment between 5 and 6 dpf.**
(MP4)

**S3 Movie. Representative movie of swimming capacity of 4-OHT-treated larvae before and during treatment.**
(MP4)

**S4 Movie. Representative movie of blood flow in the trunk circulatory system of control larvae at 6 dpf after 4-OHT treatment between 5 and 6 dpf.**
(GIF)

**S5 Movie. Representative movie of blood flow in the trunk circulatory system of atrofish larvae at 6 dpf after 4-OHT treatment between 5 and 6 dpf.**
(GIF)

**S6 Movie. Representative movie of swimming capacity of 4-OHT-treated larvae during recovery following treatment.**
(MP4)

## Acknowledgments

We thank Drs. Mosimann and Balciunas for sharing Tol2 entry vectors, p5E-*ubi:Lox-stop-Lox* and pME-*ERT2CreERT2* respectively, and Dr. Talbot for sharing the Tg(*mnx1:mCherry*) line.

# Author contributions

**Conceptualization:** Romain Menard, Cédric Dray, Jean-Philippe Pradère, Romain Madelaine.

**Data curation:** Romain Menard, Dexter Morse, Romain Madelaine.

**Formal analysis:** Romain Menard, Dexter Morse, Heath Fuqua, Joel H. Graber, James A. Coffman, Romain Madelaine.

**Funding acquisition:** Prayag Murawala, Cédric Dray, Jean-Philippe Pradère, James A. Coffman, Romain Madelaine.

**Investigation:** Romain Menard, Elena Morin, Dexter Morse, Caroline Halluin, Marko Pende, Aissette Baanannou, Janelle Grendler, Jijia Li, Laetitia Lancelot, Jessica Drent, Frédéric Bonnet, Romain Madelaine.

**Methodology:** Romain Menard, Prayag Murawala, Romain Madelaine.

**Supervision:** Romain Madelaine.

**Writing – original draft:** Romain Menard, Romain Madelaine.

**Writing – review & editing:** Romain Menard, Romain Madelaine.

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
