## [Decision Letter · Decision Letter 0]

1 Jun 2025

PGENETICS-D-25-00316

Zebrafish genetic model of neuromuscular degeneration associated with Atrogin-1 expression

PLOS Genetics

Dear Dr. Madelaine,

Thank you for submitting your manuscript to PLOS Genetics. After careful consideration, we feel that it has merit but does not fully meet PLOS Genetics's publication criteria as it currently stands. Therefore, we invite you to submit a revised version of the manuscript that addresses the points raised during the review process.

Please submit your revised manuscript within 60 days Jul 31 2025 11:59PM. If you will need more time than this to complete your revisions, please reply to this message or contact the journal office at plosgenetics@plos.org. Please include the following items when submitting your revised manuscript:

We look forward to receiving your revised manuscript.

Kind regards,

Ken M. Cadigan, PhD

Academic Editor

PLOS Genetics

Pablo Wappner

Section Editor

PLOS Genetics

Aimée Dudley

Editor-in-Chief

PLOS Genetics

Anne Goriely

Editor-in-Chief

PLOS Genetics

**Journal Requirements:**

At this stage, the following Authors/Authors require contributions: Romain Menard, Elena Morin, Dexter Morse, Caroline Halluin, Marko Pende, Aissette Baanannou, Janelle Grendler, Heath Fuqua, Jijia Li, Laetitia Lancelot, Jessica Drent, Frédéric Bonnet, Joel H. Graber, Prayag Murawala, Cédric Dray, Jean-Philippe Pradère, James A. Coffman, and Romain Madelaine. Please ensure that the full contributions of each author are acknowledged in the "Add/Edit/Remove Authors" section of our submission form.

The list of CRediT author contributions may be found here: https://journals.plos.org/plosgenetics/s/authorship#loc-author-contributions

https://journals.plos.org/plosgenetics/s/submission-guidelines#loc-parts-of-a-submission

- ® on page: 24.

5) We notice that your supplementary Figures are included in the manuscript file. Please remove them and upload them with the file type 'Supporting Information'. Please ensure that each Supporting Information file has a legend listed in the manuscript after the references list.

Potential Copyright Issues:

i) Please confirm (a) that you are the photographer of 4A, or (b) provide written permission from the photographer to publish the photo(s) under our CC BY 4.0 license.

ii) Figure S4C. Please confirm whether you drew the images / clip-art within the figure panels by hand. If you did not draw the images, please provide (a) a link to the source of the images or icons and their license / terms of use; or (b) written permission from the copyright holder to publish the images or icons under our CC BY 4.0 license. Alternatively, you may replace the images with open source alternatives. See these open source resources you may use to replace images / clip-art:

7) In the online submission form, you indicated that "All data, reagents, and genetic tools are available upon request." All PLOS journals now require all data underlying the findings described in their manuscript to be freely available to other researchers, either

1. In a public repository

2. Within the manuscript itself

3. Uploaded as supplementary information.

8) Please ensure that the funders and grant numbers match between the Financial Disclosure field and the Funding Information tab in your submission form. Note that the funders must be provided in the same order in both places as well. Currently, "3P20GM144265-01A1S1" ,"Morris Discovery Fund", "BioMaine association seed", "INSERM", "CNRS", "EFS",  and "Paul Sabatier University" are missing from the Funding Information tab.

**Reviewers' comments:**

Reviewer's Responses to Questions

Reviewer #1: This study by Menard et al. aims to provide a new model to study sarcopenia in larval zebrafish. Key findings are: (1) the development of a new gain-of-function Atrogin-1 line, specifically in skeletal muscle, (2) loss of both muscle and neurons in the model, and (3) ability of zebrafish to recover from the atrophy. The fact that overexpression of Atrogin-1 causes atrophy is perhaps the most provocative result of this study as mutants of Atrogin-1 also have muscular atrophy. That said, much of the data is primarily descriptive with little mechanistic insight provided. Overall, the problem is important, the topic interesting, but experimental support for the claims is inadequate. Specific concerns are listed below:

1) The authors nicely show that treatment with 4-OHT causes mCherry expression in the muscle fibers and when crossed to the Atrogin-1 fish, a range of muscle atrophy is observed. However, there is no confirmation that Atrogin-1 is consistently upregulated after treatment. Given the range of phenotypes (which can happen with ablation studies but also with imperfect Cre recombination), finding a way to quantify the amount of Atrogin-1 that is overexpressed is critical to evaluate the rest of the paper. This is particularly important as tamoxifen is known to have an effect on muscle atrophy and fatty degeneration in mammalian models.

2) It is not clear why overexpression of Atrogin-1 and knockout of Atrogin-1 would both cause muscular atrophy and increased apoptosis. The authors mention many times a feedback-loop component to this pathway. It would help the reader if a schematic of this feedback loop (known or proposed by the authors) was available. Further, have the authors tried using their overexpression line in the context of the published mutant? Can overexpression of Atrogin-1 rescue the initial atrophy and then additional time/increase concentration of 4-OHT show new development of atrophy? While it may be impractical to generate a double transgenic in the mutant line, a CRISPANT-based approach could get at this without the need to make multiple lines.

3) Sarcopenia is primarily induced (although there are other factors) by changes in muscle synthesis signaling pathways. This leads to increased fat infiltration which can then drive inflammation and reduce strength and activity of the muscles. In Figure 1, the RNAseq suggests inflammation is occurring as cytokines are upregulated. However, there is no further analysis to confirm that the muscle atrophy is modeling sarcopenia (i.e. fat infiltration) and not a cytokine-mediated or inflammation-based atrophy like cachexia. This is critical given the accelerated timeline of the atrophy.

4) The neuronal staining is not compelling. For example, in Figure 2, the authors show that in an atrophied-muscle section, there is a loss of signal in the zn8 staining. But zn8 staining also labels commissural fibers, outer surface of somites, some ganglia and radial glial cells. In SF3, HuC/HuD staining shows a similar absence of staining. However, there is little explanation as to why the staining in the spinal cord is completely missing in just 24-48 hours. It takes time for neurons to die, particularly neurons that are not directly in contact with muscle fibers (sensory tract, interneurons, etc). Given so much of the staining in the spinal cord is gone – could the lack of signal just be due to increased fibrosis or fat infiltration which precludes the antibody from properly staining the spinal tract? This possibility seems even more likely given that HuC/HuD neurons are intact in the sensory bundles across the lateral line, which are also in contact with the atrophied muscle. The zebrafish community has a large number of plasmids and transgenic lines that label motor neurons, interneurons, and other spinal tract cells. Although getting the transgenic overexpression system into the lines may be challenging, plasmid injection would confirm loss of cells without the use of an antibody. Alternatively, staining for neuronal death with apoptosis markers or a using a plasmid-based approach (SecA5) in conjunction with the zn8 antibody at earlier timepoints would make the loss of signal more convincing.

5) The locomotor experiments are not consistent with the staining or each other. In Figure 3, Atrofish 4-OHT have no swimming at the time of treatment (0 hr). 4-OHT takes time to absorb into the larvae and then it takes time for Cre to cause recombination. Why are the transgenic fish already not swimming at the time of 4-OHT application? Are these fish orally gavage or injected with the 4-OHT? There needs to be a timeline of imaging (phallodin staining or the line, whichever is easier) that correlates loss of locomotion with muscle atrophy, and neuronal loss. Otherwise, what is the locomotion reporting on – muscle function or loss of neurons?

6) Similarly, it doesn’t make sense that at the time of washout, the Atrofish 4-OHT swim comparably to DMSO fish. Once recombination with the Cre has occurred, it is permanently overexpressed. It would take the proliferation of cells that didn’t recombine to stop the overexpression of Atrogin-1. Other larval studies with SMA, ALS, and SCI models, show it takes >24 hr for motor neurons to regenerate in 6 dpf larvae. Even if no axons that project from the brain were damaged, without the proper CPG patterning, normal free swim would not be possible within the timeframe of 4-OHT metabolism. Do the larvae exhibit different movements as compensatory actions?

Reviewer #2: This manuscript introduces and characterizes transgenic zebrafish lines (Atrofish) that conditionally overexpress the ubiquitin ligase Atrogin-1 in muscles. The authors show that this system effectively induces muscle degeneration in larvae and adults. Their characterization suggests that inflammation is induced by muscle degeneration in Atrofish larvae, myosin light chain is affected by Atrogin-1 overexpression, muscle degeneration causes non-autonomous neuronal defects in larvae, larvae can recover from Atrogei-1-induced injury, and Atrofish can be used to cause degenerative phenotypes in adults.

This model could be used to further investigate the function of Atrogin-1, and is valuable as a model for characterizing and developing treatments for muscle degeneration. The data presented are shown clearly and, for the most part, analyzed appropriately. Additional analyses, however, would strengthen the manuscript by providing a more thorough characterization, making the model more valuable. Suggestions for further analyses are described below. Not all should be necessary for publication, but each could clarify the phenotypes observed.

Major comments:

1. Transcriptional analyses suggests an increase in inflammation in Atrofish. This is an interesting observation, and suggests that the immune system may play a role in the Atrofish phenotype. Further insight into this idea could be provided by imaging innate immune cells (macrophages and neutrophils) in relation to muscle damage. This can be done in zebrafish relatively easily with stains or immunostaining, or even better, with live imaging of macrophage reporter lines that could be followed in live animals during degeneration.

2. By staining with F310, the authors show that loss of myosin light chain accompanies muscle atrophy, and suggest that loss of myosin light chain may precede overt disruption of muscle fibers, as imaged with Phalloidin. This conclusion is based on the observation that in a small number of cases (<10% with 1-day OHT exposures, according to Figure S2C) F310 staining is impaired without obvious loss of phalloidin staining. For this argument to be more convincing, the authors should perform a time-course, which, if there conclusion is correct, may demonstrate that F310 is lost earlier after exposure than phalloidin staining is lost. (The most convincing way to test the role of myosin light chain in degeneration, which may be beyond the scope of this report, would be to determine if overexpression of the light chain in Atrofish delays muscle degeneration.)

3. The observation that motor neurons are affected non-autonomously in Atrofish is interesting, but this observation could be presented more clearly and quantified better. For example, the defects in motor neurons are difficult to see in Figure 2C with the znp-1 and zn-8 staining. Higher magnification/higher resolution images could help, as well as tracing and quantification of axon length and synapse density. Using a transgenic line that labels motor neurons clearly would be a better way to characterize motor axons. Finally, a time course quantifying axon degeneration over time after Atrogen-1 induction, relative to muscle degeneration, would be most convincing.

4. The loss of staining for neurons throughout an entire segment of the spinal cord (Fig S3C) is surprising and could be exciting, but staining with HuC/D is not the best way to appreciate this phenotype. Loss of staining for this antigen may not be equivalent to loss of neurons. Staining for cell death and/or using a transgenic line to label all neurons would provide insight. It’s hard to imagine how all the neurons in a part of the spinal cord could die, but one possibility is that inflammation in that region is killing neurons and/or impeding staining. Imaging macrophages and microglia, as suggested in point 1 above, could address this possibility.

5. The recovery experiments are surprising. Data previously presented in the manuscript shows that 5-OHT treatment, even for 24 hours, leads to cellular damage of muscles and neurons. Yet, if I understand the data correctly, immediately after removing fish from 5-OHT they behave completely normally (Figure 3B). This does not seem likely to be enough time to repair cellular damage (i.e., activate stem cells to make new muscle fibers, regrow MN axons and synapses). Transcriptional analyses are insufficient to characterize this “recovery” process. Staining for muscle and motorneurons at mulutiple timepoints is necessary to understand this process. The best way to do this experiment would be with a transgenic reporter that labels muscles, so that atrophy and recovery could be monitored longitudinally in live fish.

Minor comment:

1. For the analysis of muscle fibers presented in Figure 1B, the specific criteria distinguishing mild and strong phenotypes should be described.

**Have all data underlying the figures and results presented in the manuscript been provided?**

Reviewer #1: **No:** I may have missed the excel sheets for all the numerical data but I did not see them attached

Reviewer #2: Yes

PLOS authors have the option to publish the peer review history of their article (what does this mean? ). If published, this will include your full peer review and any attached files.

**Do you want your identity to be public for this peer review?** For information about this choice, including consent withdrawal, please see our Privacy Policy .

Reviewer #1: **Yes:** Karen Mruk

Reviewer #2: No

**Figure resubmission:**
---

## [Decision Letter · Decision Letter 1]

19 Nov 2025

PGENETICS-D-25-00316R1

Zebrafish genetic model of neuromuscular degeneration associated with Atrogin-1 expression

PLOS Genetics

Dear Dr. Madelaine,

Thank you for submitting your manuscript to PLOS Genetics. After careful consideration, we feel that it has merit but does not fully meet PLOS Genetics's publication criteria as it currently stands. Therefore, we invite you to submit a revised version of the manuscript that addresses the points raised during the review process.

We look forward to receiving your revised manuscript.

Kind regards,

Ken M. Cadigan, PhD

Academic Editor

PLOS Genetics

Pablo Wappner

Section Editor

PLOS Genetics

Aimée Dudley

Editor-in-Chief

PLOS Genetics

Anne Goriely

Editor-in-Chief

PLOS Genetics

**Additional Editor Comments:**

Considering the comments of reviewer 2, you do not have to address the time course that they propose (comment #2). But I think comments 1 and 3 are important - including controls that 4-OHT and DMSO by themselves or in combination do not affect your data seems an important point, especially regarding your surprising recovery data. Please also consider comment #4 and whether removing the Zn8 would be acceptable.

**Reviewers' comments:**

Reviewer's Responses to Questions

**Comments to the Authors:**

Reviewer #1: The authors present a new genetic model for muscular atrophy using overexpression of atrogin-1 as a driver for accelerated muscular atrophy. The authors have added new live imaging studies which strengthen their conclusions. I am in favor of publication.

Reviewer #2: This revised manuscript describes a zebrafish model of Atrogen-mediated muscle degeneration. The authors have revised the manuscript with additional experiments, analyses, and text revisions. The authors have not been able to answer all reviewer comments, but they have carried out the most feasible suggestions and met the reviewer comments halfway.

Key revisions that have improved the manuscript include better quantification of Atrogin-1 expression level increases in Atrofish, staining for fat infiltration, using a motor neuron transgenic line to better appreciate motor neuron degeneration, and using a transgenic line and in situ hybridization to assess the role of macrophages in inflammation. In particular, the use of the motor neuron line has improved the characterization of the motor neuron degeneration phenotype.

Notwithstanding the improvements to the manuscript, a few concerns remain that the authors should address more directly, at least with additional discussion.

1. The explanation for the quick recovery of swimming behavior after 4-OHT removal is unsatisfying. If it is not a regenerative response that repairs damage, then how would this work? Is it that the 4-OHT is toxic and somehow synergizes with degeneration in Atrofish?

The explanation for this result in the text is inadequate. Perhaps I’m missing something, but I do not understand what the authors mean when they write that “preventing atrogin-1 overexpression [presumably by removing 4-OHT] in muscle fibers allows for an efficient return to a homeostatic and functional state.” How would removing 4-OHT prevent overexpression? Presumably the Cre recombination has already taken place and overexpression is continuing.

Since this is a confusing section of the manuscript, the authors should discuss a plausible explanation for interpreting this “recovery” result. In the absence of a feasible interpretation, this finding is not meaningful and could be misleading--if that is the case, the authors should consider removing it from the manuscript.

2. The evidence suggesting that loss of mlc precedes muscle degeneration remains relatively weak without a time course experiment. I appreciate that a longitudinal experiment is not feasible to do in live animals, but the authors could do a time-course by staining batches of animals at different time-points after induction to determine if there are early time points when there are more mlc-negative, intact muscles. Alternatively, since this isn’t a critical point in the paper, it would be acceptable for the authors to discuss this result more equivocally.

3. The authors sometimes appear to be using 4-OHT in wildtype animals as a control, sometimes use DMSO with Atrofish as a control, and sometimes use both controls. It is not clear why they chose to use different controls in different experiments, but the results are strongest when both controls are used. In particular, it would be reassuring to see the 4-OHT/WT control for the motor neuron/spinal cord degeneration experiments, rather than just the DMSO/Atrofish control. This would rule out the possibility that the hormone is detrimental to neurons.

4. It is not clear how to reconcile the data in S2A showing no changes in macrophages in the transgenic line with the HCR data showing an increase in the macrophage transcript mpeg (Figure 3C). This discrepancy should be discussed.

5. The Zn8 staining in Figure 2 is very grainy and looks like much of it may be background staining. This staining is less clear than zn8 staining I’ve seen in other papers, suggesting that there may be a problem with the authors’ batch of antibody. Since the znp-1 staining and transgenically labeled motor neurons make the same point, this zn8 staining adds little to the manuscript and could probably be removed.

Minor point:

1. What are the yellow blobs in the 3D reconstructions shown in Figure 3B? They are not explained in the text or figure legend.

**Have all data underlying the figures and results presented in the manuscript been provided?**

Reviewer #1: Yes

Reviewer #2: Yes

PLOS authors have the option to publish the peer review history of their article (what does this mean? ). If published, this will include your full peer review and any attached files.

**Do you want your identity to be public for this peer review?** For information about this choice, including consent withdrawal, please see our Privacy Policy .

Reviewer #1: **Yes:** Karen Mruk

Reviewer #2: No

**Figure resubmission:**
---

## [Editor Report · Decision Letter 2]

31 Dec 2025

Dear Dr Madelaine,

We are pleased to inform you that your manuscript entitled "Zebrafish genetic model of neuromuscular degeneration associated with Atrogin-1 expression" has been editorially accepted for publication in PLOS Genetics. Congratulations!

Yours sincerely,

Ken M. Cadigan, PhD

Academic Editor

PLOS Genetics

Pablo Wappner

Section Editor

PLOS Genetics

Aimée Dudley

Editor-in-Chief

PLOS Genetics

Anne Goriely

Editor-in-Chief

PLOS Genetics

BlueSky: @plos.bsky.social

Comments from the reviewers (if applicable):

**Data Deposition**

http://datadryad.org/submit?journalID=pgenetics&manu=PGENETICS-D-25-00316R2

**Press Queries**

---

## [Editor Report · Acceptance letter]

PGENETICS-D-25-00316R2

Zebrafish genetic model of neuromuscular degeneration associated with Atrogin-1 expression

Dear Dr Madelaine,

We are pleased to inform you that your manuscript entitled "Zebrafish genetic model of neuromuscular degeneration associated with Atrogin-1 expression" has been formally accepted for publication in PLOS Genetics! Your manuscript is now with our production department and you will be notified of the publication date in due course.

With kind regards,

Anita Estes

PLOS Genetics

On behalf of:
